# Denitrification in soil as a function of oxygen availability at the microscale

Lena Rohe[1], Bernd Apelt[1], Hans-Jörg Vogel[1], Reinhard Well[2], Gi-Mick Wu[3], Steffen Schlüter[1]

[1]Helmholtz Centre for Environmental Research – UFZ, Department Soil System Sciences, Theodor-Lieser Str. 4, 06120 Halle, Germany

[2]Thünen Institute of Climate Smart Agriculture, Bundesallee 65, 38116 Braunschweig, Germany

[3]Helmholtz Centre for Environmental Research – UFZ, PACE, Permoserstraße 15, 04318 Leipzig, Germany

*Correspondence to*: Lena Rohe, (lena.rohe@ufz.de)

**Abstract**

The prediction of nitrous oxide ($N_2O$) and of dinitrogen ($N_2$) emissions formed by biotic denitrification in soil is notoriously

difficult, due to challenges in capturing co-occurring processes at microscopic scales. $N_2O$ production and reduction depend on the spatial extent of anoxic conditions in soil, which in turn are a function of oxygen ($O_2$) supply through diffusion and $O_2$ demand by respiration in the presence of an alternative electron acceptor (e.g. nitrate).

This study aimed to explore controlling factors of complete denitrification in terms of $N_2O$ and ($N_2O+N_2$) fluxes in repacked soils by taking micro-environmental conditions directly into account. This was achieved by measuring micro-scale oxygen

saturation and estimating the anaerobic soil volume fraction (*ansvf*) based on internal air distribution measured with X-ray computed tomography (X-ray CT). $O_2$ supply and demand was explored systemically in a full factorial design with soil organic matter (SOM, 1.2 and 4.5%), aggregate size (2-4 and 4-8 mm) and water saturation (70, 83 and 95% WHC) as factors. $CO_2$ and $N_2O$ emissions were monitored with gas chromatography. The $^{15}N$ gas flux method was used to estimate the $N_2O$ reduction to $N_2$.

N-gas emissions could only be predicted well, when explanatory variables for $O_2$ demand and $O_2$ supply were considered jointly. Combining $CO_2$ emission and *ansvf* as proxies of $O_2$ demand and supply resulted in 83% explained variability in ($N_2O+N_2$) emissions and together with the denitrification product ratio [$N_2O/(N_2O+N_2)$] (*pr*) 81% in $N_2O$ emissions. $O_2$ concentration measured by microsensors was a poor predictor due to the variability in $O_2$ over small distances combined with the small measurement volume of the microsensors. The substitution of predictors by independent, readily available proxies for $O_2$ demand

(SOM) and $O_2$ supply (diffusivity) reduced the predictive power considerably (60% and 66% for $N_2O$ and ($N_2O+N_2$) fluxes, respectively).

The new approach of using X-ray CT imaging analysis to directly quantify soil structure in terms of *ansvf* in combination with $N_2O$ and ($N_2O+N_2$) flux measurements opens up new perspectives to estimate complete denitrification in soil. This will also contribute to improving $N_2O$ flux models and can help to develop mitigation strategies for $N_2O$ fluxes and improve N use

efficiency.

Keywords: anaerobic soil volume fraction, air distance, diffusivity, nitrous oxide, dinitrogen, oxygen microsensors, product ratio, X-Ray computed tomography (X-ray CT)


## 1. Introduction

Predicting emissions of the greenhouse gas nitrous oxide ($N_2O$) is important in order to develop mitigation strategies. Agriculture accounts for approximately 60% of anthropogenic $N_2O$ emissions, most likely because high amounts of substrates for $N_2O$ producing processes result from nitrogen (N) fertilization on agricultural fields (Syakila and Kroeze, 2011; Thompson et al., 2019; Tian et al., 2020). The required process understanding is hindered, since various microbial species are capable of $N_2O$ production via several pathways and these may co-exist due to different micro-environmental conditions within short distances in soil (Hayatsu et al., 2008; Braker and Conrad, 2011). Denitrification is one of the major biological pathways for $N_2O$ production, which describes the reduction of nitrate ($NO_3^-$) as the alternative electron acceptor into the trace gas nitrous oxide ($N_2O$) as an intermediate and molecular nitrogen ($N_2$) as the final product (Knowles, 1982; Philippot et al., 2007). Although it is well known that not all microbial species are capable of denitrification pathway, it is particularly widespread among bacteria, but also several fungi and even archaea can denitrify (Shoun et al., 1992; Cabello et al., 2004).

$N_2O$ emissions from soils are often considered to be erratic in nature due to their high variability in space and time (Butterbach-Bahl et al., 2013). The low predictability is caused by the mechanisms that regulate microbial denitrification at the pore scale which are concealed from measurement techniques that average across larger soil volumes. This experimental study is designed to reveal the drivers of oxygen ($O_2$) supply and demand at the microscale that govern microbial denitrification at the macroscale.

In general, there are several controlling factors for microbial denitrification in soil. Proximal factors, such as N and carbon (C) are needed to ensure the presence of electron acceptors and electron supply. In addition, the absence of oxygen is required to express the enzymes for the reduction of reactive nitrogen. Distal factors, i.e. physical and biological factors like soil structure, soil texture, pH or microbial community, on the other hand affect the proximal factors (Groffman and Tiedje, 1988; Tiedje, 1988). The main physical controlling factors that regulate $O_2$ supply are water saturation and soil structure, because they determine the pathways through which gaseous and dissolved oxygen, but also $NO_3^-$ and dissolved organic matter may diffuse towards the location of their consumption. Likewise they determine the pathways through which denitrification products may diffuse away from these locations. In addition, both, saturation and soil structure, contribute to the regulation of $O_2$ demand through their impact on substrate accessibility and thus microbial activity (Keiluweit et al., 2016). Studies have shown microbial activity, described by microbial respiration, to increase with increasing water saturation, but it also decreased when water saturation exceeded a certain optimal value at intermediate conditions (Davidson et al., 2000; Reichstein and Beer, 2008; Moyano et al., 2012). Low water saturation causes C substrate limitations whereas high water saturation causes limited oxygen diffusion (Davidson et al., 2000). This observation goes along with an increase of anaerobic respiration in microbial hot spots when $O_2$ demand exceeded $O_2$ supply and denitrification is favoured (Balaine et al., 2015).

These physical processes that govern denitrification at the microscale have to be effectively described by macroscopic bulk soil properties in order to improve the predictability of denitrification activity at larger scales. It has been shown repeatedly that soil diffusivity can be used to predict the impact of $O_2$ supply on $N_2O$ and $N_2$ emissions (Andersen and Petersen, 2009; Balaine et al., 2016). First $N_2O$ emissions increase with decreasing diffusivity, but then it dramatically decreases due to $N_2$ production when diffusivity is extremely low.

Diffusivity is not routinely measured in denitrification studies as it is more difficult to measure than air content or water saturation, but there are many empirical models to estimate diffusivity based on air filled pore volume (Millington and Quirk, 1960; Millington and Quirk, 1961; Moldrup et al., 1999; Deepagoda et al., 2011). All of these metrics are only indirect metrics of the anaerobic soil volume fraction (*ansvf*) as direct measurements are difficult to obtain. Either it is measured locally via oxygen sensors with needle-type microsensors (Sexstone et al., 1985; Højberg et al., 1994; Elberling et al., 2011) or with foils (Elberling

et al., 2011; Keiluweit et al., 2018), which requires to average or to extrapolate measured $O_2$ saturation for the entire soil volume. Or it is estimated for the entire sample volume from pore distances in X-ray CT images of soil structure assuming that there is a direct relationship between pore distances and anaerobiosis (Rabot et al., 2015; Kravchenko et al., 2018).

Completeness of denitrification is another important controlling factor that modulates the relationship between $O_2$ availability and $N_2O$ emissions (Morley et al., 2014) which has previously been neglected in similar incubation studies (Rabot et al., 2015; Porre et al., 2016; Kravchenko et al., 2018). Since the $N_2$ background of air (78%) is very high, direct $N_2$ measurement from denitrification in soil is very challenging (Groffman et al., 2006; Mathieu et al., 2006). The [15]N labelling technique is a method successfully applied to determine $N_2O$ and also $N_2$ production from denitrification from [15]N amended electron acceptors ($NO_3^-$) (Mathieu et al., 2006; Scheer et al., 2020). Complete denitrification generates $N_2$ as the final product although it is assumed that 30% of denitrifying organisms lack the $N_2O$ reductase (Zumft, 1997; Jones et al., 2008; Braker and Conrad, 2011). Thus the denitrification product ratio $[N_2O/(N_2O+N_2)]$ (*pr*) was found to be very variable in soil studies covering the whole range between 0 and 1 (Senbayram et al., 2012; Buchen et al., 2016). Decreasing *pr*, i.e. relative increasing $N_2$ fraction compared to that of $N_2O$, were found with lower oxygen availability in consequence of higher water saturations and denitrification activities in soil (van Cleemput, 1998).

In this paper, we will reconcile all these metrics, i.e. soil structure, bulk respiration, diffusivity, $O_2$ distribution, *ansvf* and *pr* to assess their suitability to predict denitrification activity. This requires well defined laboratory experiments that either control or directly measure important distal controlling factors of denitrification activity like microbial activity, anaerobic soil volume and denitrification completeness.

To this end the current study presents a comprehensive experimental setup with well-defined experimental conditions but also micro-scale measurements of oxygen concentrations, soil structure and the air and water distribution at the pore scale. The [15]N tracer application was used to estimate the $N_2O$ reduction to $N_2$ and the $N_2O$ fraction originating from denitrification. To our knowledge this is the first experimental setup analyzing $N_2O$ and ($N_2O+N_2$) fluxes in combination with X-ray CT derived structure. Other important factors controlling denitrification like temperature, pH, nitrate limitation, saturation changes, microbial community structure, or plant-soil interactions were either controlled or excluded in this study.

The general objective of the present study is to systematically explore bulk respiration and denitrification as a function of $O_2$ supply and demand in repacked soils under static hydraulic conditions. $O_2$ demand was controlled by incubating soils with different soil organic matter (SOM) content. $O_2$ supply was controlled by different water saturations and different aggregate sizes. A novel approach is explored to assess microscopic $O_2$ supply directly from *ansvf* estimates based on the distribution and continuity of air-filled pores within the wet soil matrix.

We hypothesize that the combination of at least one proxy for $O_2$ supply (e.g. *ansvf*, diffusivity, air content) and one for $O_2$ demand ($CO_2$ production) is required to predict complete denitrification ($N_2O+N_2$), whereas *pr* as a proxy for denitrification completeness is required in addition to predict a single component ($N_2O$). The specific aims of our study were a) to investigate the potential of microscopic metrics for $O_2$ supply such as *ansvf* to predict complete denitrification activity and b) to explore as to how far a substitution of these predictors by classical, averaged soil properties required for larger scale denitrification models is acceptable.

## 2. Materials and Methods

### 2.1 Incubation

Fine-textured topsoil material was collected from two different agricultural sites in Germany (from a depth of 10 - 20 cm in Rotthalmünster (RM) and 3 - 15 cm in Gießen (GI) as representatives for agricultural mid-European soils (Table 1). To our knowledge, $N_2O$ field measurements only exist for GI soil which amounted to $N_2O$ emissions up to approximately 160 µg $N_2O$-N $m^{-2}$ $h^{-1}$ after fertilization (Müller et al., 2004; Kammann et al., 2008; Regan et al., 2011). Denitrification potential, however, exists in both soils, as recently investigated by Malique et al. (2019) in a laboratory experiment with both soils. A higher denitrification activity with GI soil was found compared to that of RM soil (Malique et al., 2019). According to this, these soils were chosen for the contrast in properties potentially affecting denitrification and respiration (SOM contents, pH, texture, bulk density) which induces a large difference in microbial respiration and hence $O_2$ demand under identical incubation settings. The rationale was that soil texture and bulk density should mainly govern air content and thus $O_2$ supply at a certain water saturation, whereas SOM content should mainly govern microbial activity and thus $O_2$ demand. The soils were sieved (10 mm), air-dried and stored at 6°C for several months before sieving into two different aggregate size fractions in order to induce variations in $O_2$ supply: small (2-4 mm) and large (4-8 mm). Care was taken to remove free particulate organic matter (POM) like plant residues and root fragments during sieving. Other aggregate size classes were not considered, as sieving yielded in a too low amount of larger aggregates that contained too much irremovable POM, whereas smaller aggregate classes resulted in a too fragmented pore space at the chosen scan settings.

**Table 1: Basic description of soil materials used for incubation (SOM – soil organic matter).**

| Site | Land use | Soil type (WRB) | Bulk density [g/cm³] | Clay [%] | Silt [%] | SOM [%] | C:N | pH (CaCl₂) |
|---|---|---|---|---|---|---|---|---|
| Rotthalmünster (RM) | arable | Luvisol | 1.3 | 19 | 71 | 1.21 | 8.7 | 6.7 |
| Gießen (GI) | grassland | Gleysol | 1.0 | 32 | 41 | 4.46 | 10.0 | 5.7 |

The soil material was pre-incubated at 50% water holding capacity (WHC) for two weeks to induce microbial activity after the long dry spell and let the flush in carbon mineralization pass that occurs after rewetting the soil. Three different saturation treatments were prepared for subsequent incubation experiments (70%, 83% and 95% WHC) to control the $O_2$ supply and thus provoke differences in denitrification activity. A $^{15}N$ solution was prepared by mixing 99 at% $^{15}N$-$KNO_3$ (Cambridge Isotope Laboratories, Inc., Andover, MA, USA) and unlabelled $KNO_3$ (Merck, Darmstadt, Germany) to reach 50 mg N $kg^{-1}$ soil with 60 at% $^{15}N$-$KNO_3$ in each water saturation treatment. Hence, for the two higher water saturations the stock solution was more diluted in order to reach the same target concentration in the soil. In a first step the soil was adjusted to 70% WHC before packing.

This $^{15}N$-labelled soil was filled in 2 cm intervals into cylindrical PVC columns (9.4cm inner diameter x 10cm height) (Figure 1) and compacted to a target bulk density that correspond to site-specific topsoil bulk densities (Jäger et al., 2003; John et al., 2005). Packing in five vertical intervals achieved a uniform porosity across the column. However, there were inevitable porosity gradients within intervals (Figure S4) that affected the air and water distribution and thus air continuity at high water saturations. This packing resulted in 902 and 694 g dry weight of RM and GI soil, respectively. For the latter two saturation levels the rest of $NO_3^-$ solution was sprayed sequentially onto each layer after packing. The incubation of such repacked soils instead of intact soil

columns was chosen to i) systematically investigate the effect of aggregate size and to ii) guarantee thorough mixing of the [15]N tracer with the soil.

In this way, a full factorial design with twelve treatments and three factors (soil: RM, GI; aggregate size: large, small; saturation:
70, 83, 95 % WHC) were prepared in triplicates for incubation. WHC was additionally measured for both soil materials in parallel soil cores. For a better comparability with previous studies the results will be presented in terms of water-filled pore space (WFPS), which is derived from the known mass of soil and water and their respective densities. A detailed description of the experimental setup can be found in the Supplementary Material.

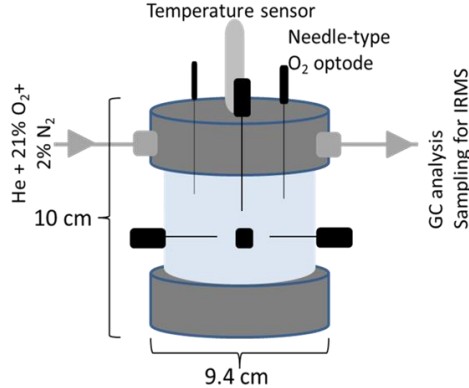


**Figure 1: Schematic of the column for repacked soil showing the dimension (10 x 9.4 cm), the lid with in- and outlet for technical gas (21 % $O_2$ and 2 % $N_2$ in helium), in black $O_2$ microsensors and in gray the temperature sensor located in soil core. The outlet of the lid was directly connected to a gas chromatography (GC) and allowed sampling for isotope ratio mass spectrometry (IRMS).**

The columns containing the packed soil aggregates were closed tightly and were equipped with an in- and outlet in the headspace (Figure 1). To analyse $O_2$ saturation, needle-type (40x0.8 mm) oxygen microsensors with <140 μm flat-broken sensor tip (NFSG-PSt1, PreSens Precision Sensing GmbH, Regensburg, Germany) were pinched through sealed holes in the lid and PVC column at seven well defined positions. Three sensors were located at the top by inserting vertically into the soil through the lid and headspace down to approximately 20 mm depth, whereas four sensors were inserted laterally at the centre of the column in
about 36 mm depth with angular intervals of 90°. The microsensors were coupled to a multi-channel oxygen meter (OXY-10 micro, PreSens Precision Sensing GmbH, Regensburg, Germany) and $O_2$ measurements were stored in 15min intervals. The $O_2$ data were aggregated to 6 hour means for further analysis. The columns were placed in a darkened, temperature-controlled 20°C water bath (JULABO GmbH, Seelbach, Germany). Two flow controllers (G040, Brooks® Instrument, Dresden, Germany) served to flush the columns with technical gas (21% $O_2$ and 2% $N_2$ in helium, Praxair, Düsseldorf, Germany) through the inlet of
the columns at a rate of 5 ml min$^{-1}$. This artificial atmosphere with low $N_2$ background concentration was used to increase sensitivity for $N_2$ fluxes (Lewicka-Szczebak et al., 2017). Initially, the headspace was flushed with technical gas for approximately 3 to 5 hours under 6 cycles of mild vacuum (max. 300mbar) to bring down the $N_2$ concentration within the soil column approximately to that of the technical gas (2%) and to ensure comparable initial conditions for incubation. Incubation time was 192 hours. Additional information on a parallel incubation where atmospheric conditions were switched from oxic to
anoxic conditions to calculate the anaerobic soil volume fraction (*ansvf$_{cal}$*) can be found in the Supplementary Material.

## 2.2 Gas analysis

*Gas chromatography (GC)*

The columns outlet was directly connected to a gas chromatograph (Shimadzu 14B) equipped with an electron capture detector (ECD) to analyse $N_2O$ and two flame ionization detectors (FID) to analyse methane (not reported) and $CO_2$. GC measurements were taken on-line every 6.5 minutes using GC Solution Software (Shimadzu, GCSolution 2.40). The detection limit was 0.25ppm $N_2O$ and 261.90ppm $CO_2$ with a precision of at least 2 and 1%, respectively. The $N_2O$ and $CO_2$ data were aggregated to 6 hour means for further analysis in order to eliminate the high frequency noise from the otherwise gradually changing gas

concentrations under static incubation conditions. The measurements during an equilibration phase of 24h were excluded. $N_2O$ fluxes derived from GC analysis may include $N_2O$ from other processes than denitrification and is thus referred as the total net $N_2O$ fluxes (*$N_2O\_total$*).

*Isotopic analysis*

Samples for isotopic analysis of $^{15}N$ in $N_2O$ and $N_2$ were taken manually after 1, 2, 4, and 8 days of incubation in 12 ml

exetainers (Labco ©Exetainer, Labco Limited, Lampeter, UK). To elute residual air from the 12 ml exetainer it was flushed three times with helium (helium 6.0, Praxair, Düsseldorf, Germany) prior evacuating the air to 180 mbar. The exetainers were flushed with headspace gas for 15min, which amounts to a six-fold gas exchange of the exetainer volume. At the end of the incubation, technical gas was also sampled to analyze the isotopic signature of the carrier gas.

These gas samples were analysed using an automated gas preparation and introduction system (GasBench II, Thermo Fisher

Scientific, Bremen, Germany, modified according to Lewicka-Szczebak et al. (2013) coupled to an isotope ratio mass spectrometer (MAT 253, Thermo Fisher Scientific, Bremen, Germany) that measured m/z 28 ($^{14}N^{14}N$), 29 ($^{14}N^{15}N$), and 30 ($^{15}N^{15}N$) of $N_2$ and simultaneously isotope ratios of $^{29}R$ ($^{29}N_2/^{28}N_2$) and $^{30}R$ ($^{30}N_2/^{28}N_2$). All three gas species ($N_2O$, ($N_2O+N_2$), and $N_2$) were analysed as $N_2$ gas after $N_2O$ reduction in a Cu oven. Details of measurement and calculations for fractions of different pools (i.e. N in $N_2O$ (*$fp\_N_2O$*) or $N_2$ (*$fp\_N_2$*) originating from $^{15}N$-labelled $NO_3^-$ pool) were described elsewhere and are

provided in Supplementary Material (Supplementary Material, Figure S3) (Spott et al., 2006; Lewicka-Szczebak et al., 2013; Buchen et al., 2016).

The product ratio (*pr*) [$N_2O/(N_2O+N_2)$] was calculated for each sample:

$$pr\ [-] = \frac{f_{p\_N_2O}}{f_{p\_N_2O}+f_{p\_N_2}} \tag{1}$$

The calculated average *pr* [$N_2O/(N_2O+N_2)$] of each treatment was also used to calculate the average total denitrification fluxes

($N_2O+N_2$ fluxes) during the incubation:

$$(N_2O + N_2)\ [\mu g\ N\ h^{-1} kg^{-1}] = \frac{N_2O\_total}{pr} \tag{2}$$

## 2.3 Microstructure analysis

Due to the experimental setup, it was only possible to scan the soil cores with X-ray CT (X-tek XTH 225, Nikon Metrology) once directly after the incubation experiment. The temperature sensor was removed, but the oxygen micro-sensors remained in

place during scanning. The scan settings (190 kV, 330 μA, 708 ms exposure time, 1.5 mm Cu filter, 2800 projections, 2 frames per projection) were kept constant for all soils and saturations. The projections were reconstructed into a 3D tomogram with 8-bit precision and a spatial resolution of 60 μm using the filtered back projection algorithm in X-tek CT-Pro. Only macropores twice this nominal resolution were clearly detectable in the soil core images. Hence, at the lowest water saturation not all air-filled pores can be resolved, which will be discussed below. The 3D images were processed with the Fiji bundle for ImageJ

(Schindelin et al., 2012) and associated plugins. The raw data were filtered with a 2D non-local means filter for noise removal. A

radial and vertical drift in grayscale intensities had to be removed (Iassonov and Tuller, 2010; Schlüter et al., 2016) before these corrected gray-scale images (Figure 2a) were segmented into multiple material classes using the histogram-based thresholding methods (Schlüter et al., 2014). The number of materials varied between two (air-filled pores, soil matrix) and four (air-filled pores, water-filled pores, soil matrix, mineral grains) depending on saturation and soil material. By means of Connected Components Labeling implemented in the MorpholibJ plugin (Legland et al., 2016) the air-filled pore space was further segmented into isolated and connected air-filled porosity, depending on whether there was a continuous path to the headspace (Figure 2b). Average oxygen supply in the core was estimated by three metrics: 1) Visible air-filled porosity ($\varepsilon_{vis}$) and connected air content ($\varepsilon_{con}$) determined by voxel counting (Figure 2b), 2) average air distance derived from the histogram of the Euclidean distances between all non-air voxels and their closest connected air voxel (Figure 2c,d) (Schlüter et al., 2019) and 3) the *ansvf* which corresponds to the volume fraction of air distance larger than a certain threshold. Therefore, in a sensitivity test, air distance thresholds of 0.6, 1.3, 2.5, 3.8 and 5.0 mm were used to estimate the *ansvf* and to find the best correlation between *ansvf* and $N_2O$ as well as ($N_2O+N_2$) fluxes. This was found with an *ansvf* at a critical air distance of 5 mm when pooling GI and RM soils (Figure 2c,d).

In summary, the $\varepsilon_{con}$ is a proxy for the supply with gaseous oxygen coming from the headspace, whereas the connected air distance and *ansvf* are proxies for the supply limitation of dissolved oxygen by diffusive flux through the wet soil matrix. In addition to these averages for entire soil cores, both $\varepsilon_{con}$ and average air distance were also computed locally in the vicinity of oxygen sensor tips (Figure 2b-c), to compare these metrics with measured oxygen concentrations. Spherical regions of interest (ROI) with different diameters from 3.6 to 10.8 mm were tested with respect to highest correlation of $\varepsilon_{con}$ and average air distance with average oxygen concentration of individual sensors. This was found to occur at a diameter of 7.2 mm, when centered on the sensor tip.

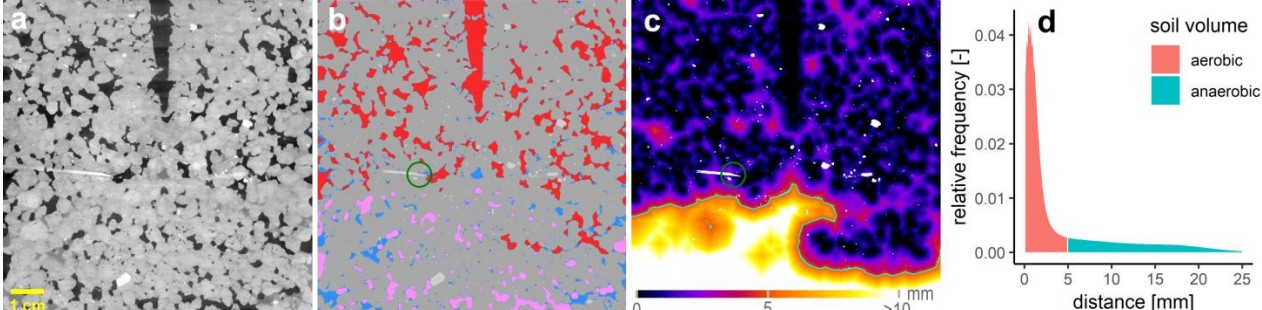

**Figure 2: 2D slice of one soil core packed with large aggregates (4-8 mm) from Gießen soil (GI) incubated at 75% WFPS to illustrate gray value contrast between materials. (a) One oxygen microsensor is shown on the left (white needle) and the hole of the temperature sensor at the top (black) within the soil matrix (gray), stones (white) and pores that are either filled with air (black) or water (light gray). (b) Material classes after segmentation including soil matrix (gray), water (blue), mineral grains (light gray), connected air (red) and isolated air (rose). The green circle around the light gray sensor tip depicts the diameter of 7.2 mm that is used to characterize its environment. (c) 3D Euclidean distance to the closest connected air voxel (mineral grains are excluded) in each soil matrix or water voxel. The closest air voxel might be outside of the 2D plane. The green line depicts the connected air distance threshold of 5 mm that differentiates between an anaerobic soil volume fraction (light colors) or aerated volume (dark colors). (d) Relative frequency of soil volume as a function of distance to closest connected air [mm] divided into aerobic (red) and anaerobic (green) soil volume.**

In addition to scans of the entire core, four individual aggregates (4-8 mm) of each soil were also scanned with X-ray CT (80 kv, 75 µA, 1s exposure time, no filter, 2400 projections, 2 frames per projection), reconstructed in 8-bit at a voxel resolution of 5 µm, filtered with a 2D non-local means filter and segmented into pores and background with the Otsu thresholding method (Otsu, 1975). The largest cuboid fully inscribed in an aggregate was cut and used for subsequent diffusion modelling as described below.

## 2.4 Diffusivity simulations

Diffusivity was simulated for individual aggregates as well as for the entire soil core (bulk diffusivity) directly on segmented X-ray CT data by solving the Laplace equation with the DiffuDict module in the GeoDict 2019 Software (Math2Market GmbH, Kaiserslautern, Germany). A hierarchical approach was used to (1) estimate the effective diffusivity of the wet soil matrix by simulating Laplace diffusion on individual soil aggregates with the Explicit Jump solver (Wiegmann and Bube, 2000; Wiegmann and Zemitis, 2006) and (2) model diffusivity ($D_{sim}$) with the Explicit Jump solver on the entire soil core (1550x1550x[1500-1600] voxels). The latter was based on the visible 3D pore space and using the effective diffusion coefficient of the soil matrix as obtained from the simulation of soil aggregates. We assumed an impermeable exterior, impermeable mineral grains (GI only) and the diffusion coefficient of oxygen in air and water (≥75%WFPS only) in the respective material classes (see detailed information in Supplementary Material).

## 2.5 Statistical analysis

Statistical analysis was conducted with R (R Core Team, 2018). Figures were produced with package ggplot2 (Wickham, 2016). In order to estimate the correlation between various variables that do not exhibit a normal distribution (average values of $N_2O$ fluxes, ($N_2O+N_2$) fluxes, $CO_2$ fluxes, $O_2$ saturation, $D_{sim}$, $\varepsilon_{con}$, *ansvf* and *pr*) Spearman's rank correlations with pairwise deletion of missing values was performed pooling data for GI and RM soils. The p-values were corrected for multiple comparison according to Benjamini and Hochberg (1995) and adjusted p-values ≤0.05 were considered as significant.

As described before, there were four missing values for *pr* due to limitation of the isotopic measurement at the lowest saturation. For further statistical analysis of the dataset, any missing *pr* values were imputed using the chained random forest using more than 100 regression trees, in terms of overall variable pattern, as this method can handle nonlinear relationships between variables (Breiman, 2001; Nengsih et al., 2019). It was also required to standardize the data of very different value ranges for further analysis. Since $N_2O$ and/or ($N_2O+N_2$) were not detectable for a few samples at the lowest saturation, a constant of 1 was added to $N_2O$ and ($N_2O+N_2$) fluxes prior transformation. This changes the mean value but not the variance of data. In order to get normal distributions and linear relationships, a logarithmic transformation was applied to metric data ($CO_2$, $N_2O$ and ($N_2O+N_2$) fluxes, $D_{sim}$), whereas a logistic transform $logit(x) = log(x/(1-x))$ was applied to dimensionless ratios between 0 and 1 (*ansvf*).

Since there was a high collinearity among most variables, a partial least square regression (PLSR) with Leave-One-Out Cross-validated $R^2$ was the best method to identify the most important independent explanatory variables (six predictors: $CO_2$ fluxes, $O_2$ saturation, $D_{sim}$, $\varepsilon_{con}$, *ansvf* and *pr*) to predict the response variables $N_2O$ or ($N_2O+N_2$) fluxes. It has to be emphasized that $N_2O$ fluxes and *pr* were measured independently of each other using different measuring methods (gas chromatography and isotopic analysis) what justifies *pr* as a predictor variable for $N_2O$ fluxes. In contrast to this ($N_2O+N_2$) fluxes were calculated from *pr* and therefore *pr* was not included in PLSR for the response variable ($N_2O+N_2$) fluxes (resulting in five explanatory variables). Bootstrapping was used to provide confidence intervals that are robust against deviations from normality (R package boot v. 1.3-24) (Davison and Hinkley, 1997; Canty and Ripley, 2019). Given the relatively small sample size (36 incubations in total), the smoothed bootstrap was used by resampling from multivariate kernel density (R package kernelboot v. 0.1.7) (Wolodzko, 2020). The BCa bootstrap confidence interval of 95% of $R^2$ was a measure to explain the variability in each response variable (Efron, 1987). Components that best explained $N_2O$ and ($N_2O+N_2$) fluxes were identified by permutation testing.

To address the second research question of this study concerning substitutions of predictors by classical, averaged soil properties additional and simplified models with the PLSR approach described above were performed using various variables to substitute

most important predictors for $N_2O$ or $(N_2O+N_2)$ fluxes. A detailed description of the substitution is provided in the result section 3.4 and discussion section 4.2.

## 3 Results

### 3.1 Bulk respiration

Time series of $CO_2$ and $N_2O$ fluxes (Supplementary Material, Figure S1) show aggregated values for six hour steps over the complete incubation time of approximately 192 hours, ignoring the first 24 hours due to initial equilibration of the system (i.e. redistribution of water, expression of all denitrification enzymes, fast mineralization of labile carbon). Averages for the whole incubation are reported in Figure 3a, 3c and in Supplementary Material, Table S1, Table S2. The 3.7 times higher SOM content in GI soil than in RM soil resulted in higher microbial activity so that $CO_2$ fluxes were approximately three times higher, for all

saturations. The variability in $CO_2$ fluxes between replicates is much higher than the temporal variability during incubation. This is probably explained by small differences in packing of the columns that can have large consequences for soil aeration. $CO_2$ production in both soils was lowest with highest water saturation but were quite similar for both treatments with saturations <80% WFPS (Figure 3a). Aggregate size had a negligible effect on $CO_2$ production.

  Substantial $N_2O$ and $(N_2O+N_2)$ emissions were detected for saturations ≥75% WFPS and were again approximately three times

higher in SOM-rich GI soil than in RM soil (Figure 3c,d). The variability between replicates is again higher than the temporal variability (e.g. in Figure 3d and time series in Supplementary Material, Figure S1) and the effect of aggregate size is inconsistent due to the large variability among replicates. Mineral N was not analyzed after the incubation and therefore cumulative $(N_2O+N_2)$ fluxes were used to estimate the N loss after 192h of incubation. Considering the N addition of 50 mg N $kg^{-1}$ as $NO_3^-$ and an average natural $NO_3^-$ background of 34 mg $kg^{-1}$ substantial N loss was observed for both soils at ≥75%

WFPS. The N converted to $N_2O$ or $N_2$ represents a proportion equal to ≤2.6% with RM soil and ≤8.0% with GI soil for both aggregate sizes and saturations.

  Average $O_2$ saturation was lowest with highest water saturation and roughly the same for saturations <80% WFPS (Figure 3b). Some sensors showed a gradual decline in $O_2$ concentration, whereas some showed a drastic reduction or increase in a short period of time, probably due to water redistribution (Supplementary Material, Figure S2). The average of the final 24h was taken

for all subsequent analysis, as this probably best reflects the water distribution scanned with X-ray CT. Standard errors among the seven $O_2$ microsensors were high in each treatment due to very local measurement of $O_2$ that probed very different locations in the heterogeneous pore structure.

  The *pr*, i.e. the $N_2O/(N_2O+N_2)$ as a measure of denitrification completeness, showed a similar behavior as a function of water saturation like $N_2O$ release with a plateau for saturations ≥75% WFPS at 0.6 and a lower, but somewhat more erratic *pr* for the

lowest saturation due to a generally low $^{15}N$ gas release (Figure 3e). Thus, the $(N_2O+N_2)$ fluxes at ≤65% WFPS could only be calculated for a small number of samples, due to lacking data of *pr* (Supplementary Material, Table S1, Table S4). SOM content and aggregate size had no effect on *pr*. Time series of *pr* showed a gradual reduction for all treatments as the $N_2$ emissions grew faster than the $N_2O$ emissions (Supplementary Material, Figure S5). With water saturations >75% WFPS the *pr* decreased with time and was in most cases <0.5 at the end of incubation (Supplementary Material, Figure S5). In summary, for each soil all

samples with saturation ≥75% WFPS showed similar *pr* (Figure 3e) and $N_2O$ release (Figure 3c). This agreed well with subsequent X-ray CT estimates of air connectivity as shown below.

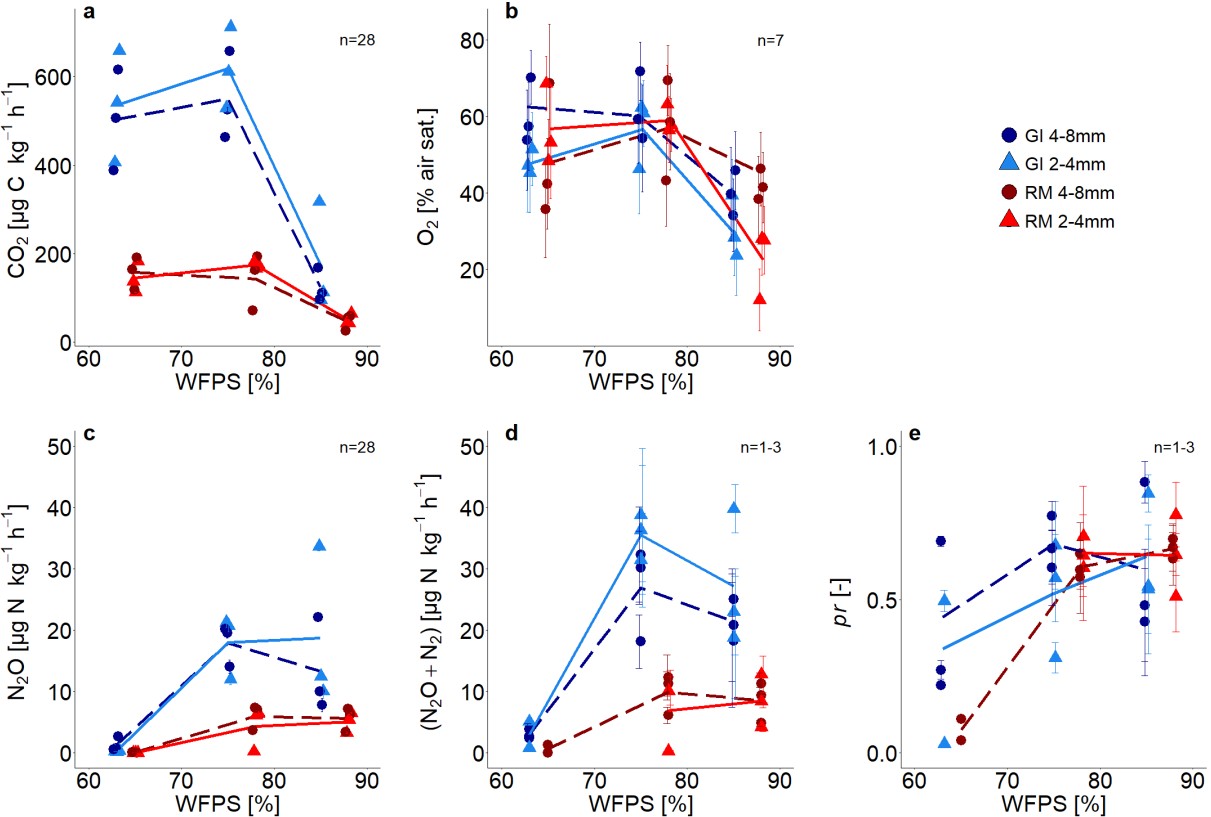

**Figure 3: (a)** Average $CO_2$ fluxes, **(b)** average $O_2$ saturation, **(c)** average $N_2O$ and **(d)** $(N_2O+ N_2)$ fluxes and **(e)** average product ratio *(pr)* $[N_2O/(N_2O+N_2)]$ as a function of water filled pore space (WFPS) for two repacked aggregate sizes (2-4 and 4-8 mm) from Rotthalmünster (RM) and Gießen (GI) soil. Symbols depict the average values for each of three individual replicates with error bars showing the standard error of the mean; standard error in (a) and (c) of fluxes measured during incubation, in (b) the standard error from measurements of seven sensors located within the soil core and in (d) and (e) of three measurements during incubation time (after 2, 4, and 8 days with detectable R[29] and R[30]; n= 3 for two highest WFPS). The number of measurements (n) considered for averaging are displayed in each subfigure. The lines (dashed and solid) connect the average value of three replicates at each saturation (large and small aggregates, respectively).

## 3.2 Pore system of soil cores

Due to lower target bulk density in GI soil (1.0 g cm$^{-3}$) compared to that of RM soil (1.3 g cm$^{-3}$) visible air content ($\varepsilon_{vis}$, depicted in red and pink in Figure 2c) was higher independent of aggregate size (Figure 4a). The $\varepsilon_{vis}$ decreased with increasing water saturation, but not linearly as would be expected. The air contents in the very wet range are in fact higher (16-17%), than the target air saturation of approximately 11 or 15% for RM and GI soil, respectively. It was not possible to remove air more efficiently during packing and some ponding water might have accidentally been removed with vacuum application during purging at the beginning of incubation. Additionally, the GI soil was rich in vermiculite and swelled upon wetting. This increase in soil volume at the end of incubation resulted in a relative decline in water content. For increasing water content the air content that is connected to the headspace ($\varepsilon_{con}$, depicted in red in Figure 2c) was reduced much more strongly as compared to the total $\varepsilon_{vis}$. This was observed for both soils and aggregate sizes and indicates that, a substantial amount of air is trapped (Figure 4b). According to this observation, average distance to visible air was very small (Figure 4c) and remained below 1.5 mm even for the highest water saturation with generally smaller distances for smaller aggregates. Yet, the average distance to the pore system connected with headspace escalates in the wet range (Figure 4d). The huge variability among replicates comes from the fact that trapping by complete water blockage typically occurs in the slightly compacted upper part of a packing interval, but the specific interval where this happens varies among samples (Supplementary Material, Figure S4). The different aggregate sizes did not

affect the distance to connected air as the long-range continuity of air is controlled by bottle-necks in the pore space and not by aggregate size.

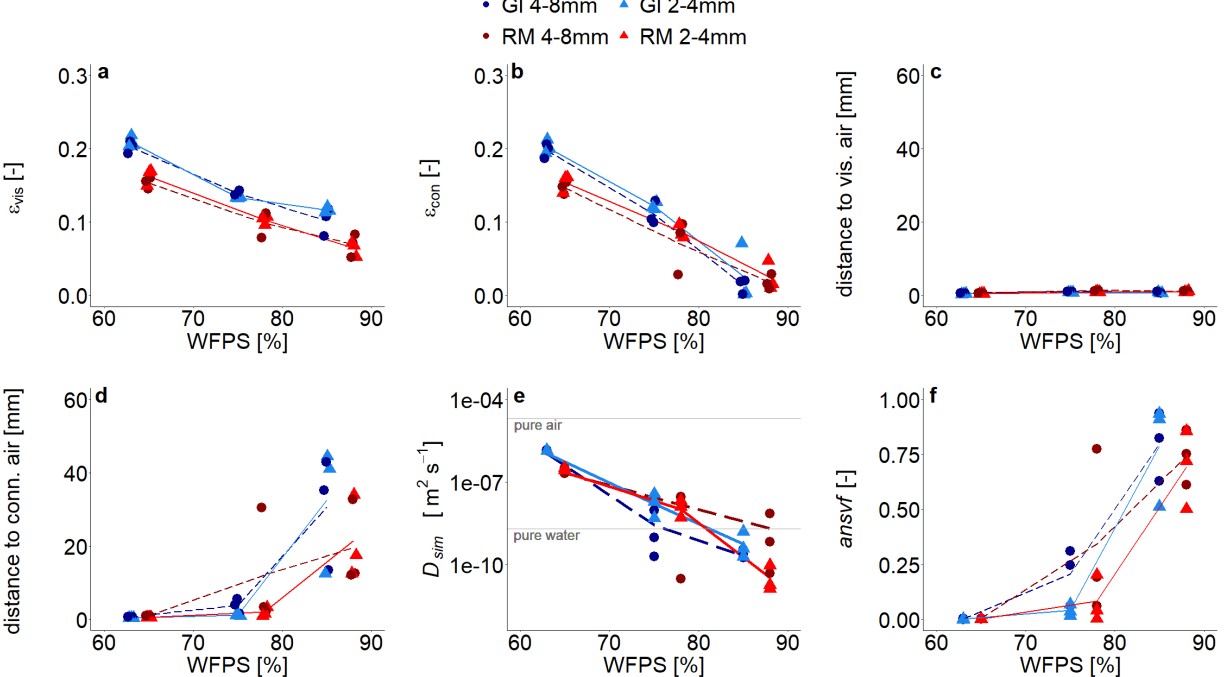

**Figure 4: (a) Visible air content ($\varepsilon_{vis}$), (b) connected air content ($\varepsilon_{con}$), (c) average distance to visible air, (d) average distance to connected visible air, (e) simulated diffusivity ($D_{sim}$) and (f) anaerobic soil volume fraction (*ansvf*) as a function of water filled pore space (WFPS) for two repacked aggregate sizes (2-4 and 4-8 mm) from Rotthalmünster (RM) and Gießen (GI) soil, and three replicates each depicted by symbols. The lines (dashed and solid) connect the average value of three replicates (large and small aggregates, respectively). The horizontal gray lines in (e) reflect material properties. The experiment was performed at 20°C and**
**according to that diffusivity was calculated at 20°C.**

Water saturation had a dramatic impact on $D_{sim}$ (Figure 4e) leading to a reduction by five orders of magnitude in a rather small saturation range. At high saturations it fell below the oxygen diffusion coefficient in pure water due to the tortuosity of the pore system. The *ansvf* (Figure 4f) is directly linked to connected air distance and shows the same escalating behavior at the highest

saturation up to a volume fraction of 50-90%. The *ansvf* is highly correlated with $CO_2$ emissions (Spearman's $R>-0.7$ and p=0.04) which exhibits the same tipping point behavior, yet with very different slopes in the regression for the different soils due to different microbial activity (Figure S6). The correlation of *ansvf* is weaker with $N_2O$ (Spearman's $0.6<R<0.77$, p<0.1) and negligible with ($N_2O+N_2$) (p>0.2), suggesting that denitrification is more complexly controlled. The full regression analysis of *ansvf* with different gases and for different soils and aggregate sizes is presented in the supporting information (Figure S6).


### 3.3 Microscopic oxygen distribution

The local measurements of $O_2$ using microsensors is demonstrated as an example for two selected sensors from the same soil column (GI soil incubated at 75% WFPS). They are located in the same depth with a separation distance of <2 cm. Sensor 1 detected low $O_2$ concentrations (18% air saturation) because it was located in a compact area with low $\varepsilon_{con}$ (4%) and a rather

large distance to the closest air-filled pore (1.6 mm) (Figure 5a,b,d). Sensor 2 detected fairly high $O_2$ concentrations (76% air saturation) as it happened to pinch into a macropore with a high $\varepsilon_{con}$ (15%) and a short distance to connected air (0.8 mm) in its vicinity (Figure 5a-c). The green or violet circle with a diameter of 7.2 mm depicts the spherical averaging volume for $\varepsilon_{con}$ and

distance to connected air that correlated best with the average $O_2$ concentrations when lumped over all soils and saturations (Figure 5b-d).

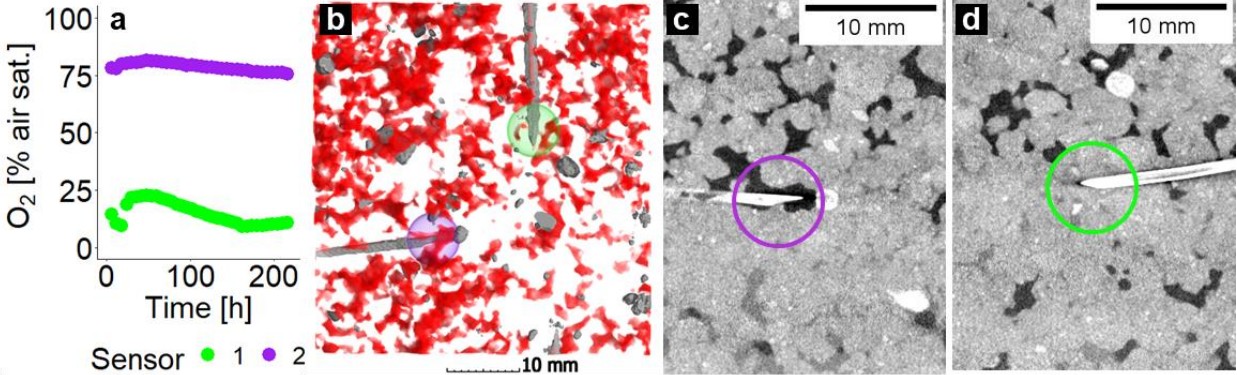


**Figure 5: Local oxygen distribution in one soil core packed with small aggregates (2-4 mm) from Gießen soil (GI) incubated at 75% water filled pore space (WFPS) to illustrate as an example the very local measurement of $O_2$. Shown here are (a) $O_2$ saturations measured by two microsensors as a function of incubation time, (b) a 3D subvolume shown from the top with connected air depicted in red and both sensors depicted with their respective spherical support volume in colors corresponding to (a), and 2D gray scale slices**
**through the sensor tip depicting soil matrix in light gray, water in dark grey, and air in black for(c) the sensor measuring high and for (d) the sensor measuring low $O_2$ saturations. The violet or green circles depict the proximity of the sensor tip (7.2 mm diameter) used to calculate the averaged local metrics.**

The treatment specific correlations between distance to connected air and average $O_2$ concentrations are shown in Figure 6. At the lowest saturation level there is no correlation at all (Spearman's $-0.4 << R < 0.1$ and p $\geq 0.38$, Figure 6a,d), because some
unresolved pores ($< 120$ µm) within the aggregates are air-filled so that oxygen availability is not limited by visible air. At the intermediate saturation level the correlations were best (Spearman's $R < -0.7$ and p$\leq 0.02$) because all unresolved pores are water-filled (Figure 6b,e). At the highest water saturation the correlation was highest for large aggregates (Spearman's $R = -0.6$ and p $= 0.08$), because the local effect of soil structure might become stronger relative to the non-local effect of air entrapment. With the other three treatments the correlation were worse again (Spearman's $R$ between -0.01 and -0.3 and p$\geq 0.58$, Figure 6c,f),
because distance to connected air ignores all trapped air which may still contribute a lot to oxygen supply.

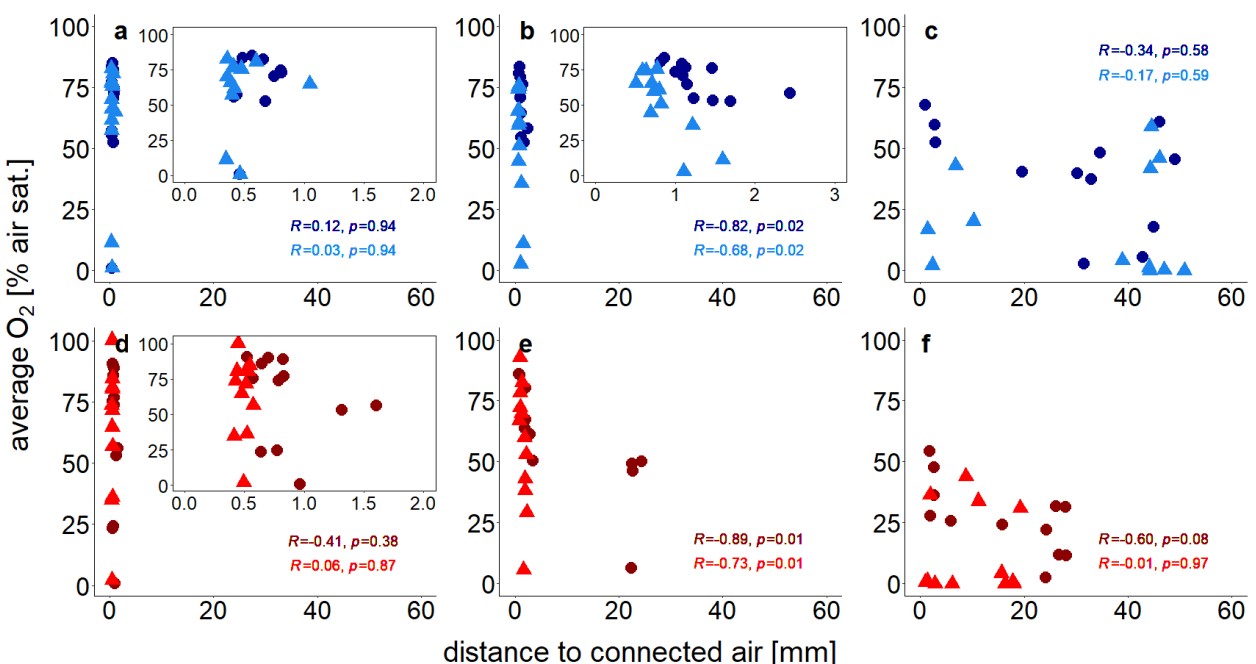

**Figure 6:** Average $O_2$ saturation (at the end of incubation experiment) measured with four sensors each located at the center of soil core as a function of distance to visible connected air for two repacked aggregate sizes (2-4 mm and 4-8 mm) from Gießen (GI, (a)-(c), blue) and Rotthalmünster (RM, (d)-(f), red) soil. (a) and (d) show results for lowest (63 or 65 % water filled pore space (WFPS) with GI and RM soil, respectively), (b) and (e) for medium (75 or 78 % WFPS with GI and RM soil, respectively), and (c) and (f) for highest (85 or 88 % WFPS with GI and RM soil, respectively) water saturation. The insets in (a), (b), and (d) show a reduced distance range. The distance to visible connected air is averaged in a spherical region around the sensor tip (7.2 mm diameter). The Spearman's rank correlation coefficient (*R*) indicate the extent of monotonic relation between the ranks of both variables. The associated p-values (*p*) were corrected for multiple comparison according to Benjamini and Hochberg (1995).

### 3.4 Explanatory variables for denitrification

So far the correlations among different explanatory variables and between explanatory variables and N-gas release have been shown for individual treatments, i.e. separately for each combination of soil and aggregate size, in order to focus on the effect of water saturation. However, the true potential of explanatory variables to predict denitrification can only be explored with the entire pooled data set, so that the variability in denitrification is captured more representatively.

The PLSR identified two principal components that best explained $N_2O$ and $N_2O+N_2$ fluxes, while most variables contributed to the first component (Comp1) and almost exclusively $CO_2$ release contributed to the second component (Comp2) (see Supplementary Material S8). These principal components revealed vastly different ability of individual explanatory variables to explain the observed variability in $N_2O$ and ($N_2O+N_2$) release. The importance of explanatory variables to predict $N_2O$ and $N_2O+N_2$ fluxes varied as follows: $CO_2$ > (*pr* >) *ansvf* > $D_{sim}$ > $\varepsilon_{con}$ > $O_2$ (see Supplementary Material Figure S8). Hereinafter *pr* shown in brackets illustrates its contribution to PLSR analysis for $N_2O$ fluxes only. The explanatory variability, expressed in the text as $R^2*100$ [%], was 82% for $N_2O$ fluxes and 78% for $N_2O+N_2$ fluxes when considering the complex model with all explanatory variables ($CO_2$ flux, $O_2$ saturation, $\varepsilon_{con}$, $D_{sim}$, *ansvf* (and *pr*)) (Figure 7). The resulting regression equations can be found in Supplementary Material (Equation 7-8).

Starting from this complex model a series of simplifications and substitutions of explanatory variables was conducted to assess in how far the resulting loss in predictive power is acceptable. Reducing the number of explanatory variables to the most important variables resulted in $CO_2$ and *ansvf* for ($N_2O+N_2$) release (83% explained variability, simplified model in Figure 8). In other

words, the combination of these two predictors (*ansvf* and $CO_2$) is crucial, as $CO_2$ release explains the different denitrification

rates between the two soils, whereas *ansvf* explains the differences within a soil due to different saturations. To predict $N_2O$ emissions the simplified model with most important explanatory variables $CO_2$, *ansvf* and *pr* as a third predictor resulted in 81% of explained variability (Figure 8). Average $O_2$ saturation could be omitted for its small correlation with $N_2O$ or ($N_2O+N_2$) release in general, whereas $\varepsilon_{con}$ and $D_{sim}$ could be omitted because of the high correlation with *ansvf* (Supplementary Material, Figure S7).

The regression equations with $R^2$ values and a confidence interval of 95% in square brackets resulting from PLSR with $CO_2$, *ansvf* (and *pr*) identified as most important explanatory variables to predict $N_2O$ or ($N_2O+N_2$) fluxes of the present study for data after log- or logit transformation:

$$\log(N_2O) = \ 0.65 \ \log(CO_2) + 0.74 \ \text{logit}(ansvf) + 0.75 \ pr; R^2 = 0.81 \ [0.67\text{-}0.89] \tag{3}$$

$$\log(N_2O + N_2) = 1.14 \ \log(CO_2) + \ 1.60 \ \text{logit}(ansvf) \ ; R^2 = 0.83 \ [0.71\text{-}0.90] \tag{4}$$

Various variables were used to substitute best predictors ($CO_2$ or *ansvf*) (Figure 7) in PLSR. The substitution of $CO_2$ by SOM or *ansvf* by $\varepsilon_t$, $D_{sim}$ or empirical diffusivity ($D_{emp}$) based on total porosity and air content (Deepagoda et al., 2011) is explained in the discussion section 4.2.

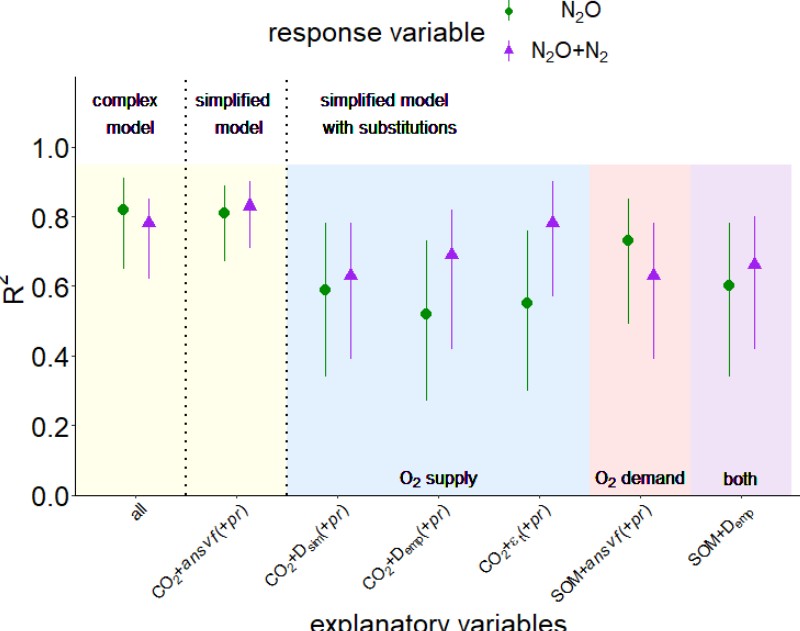

Figure 7: Explained variability expressed as $R^2$ with a confidence interval of 95% resulting from partial least square regression (PLSR) with Leave-One-Out Cross-validation and bootstrapping for response variables $N_2O$ (green symbols) or ($N_2O+N_2$) fluxes (violet symbols) for pooled data of both soils (from Rotthalmünster (RM) and Gießen (GI)), WFPS treatments and aggregate sizes (n= 36). The yellow area shows a complex model including all explanatory variables of the present study ($CO_2$, $O_2$, connected air content ($\varepsilon_{con}$), diffusivity ($D_{sim}$), anaerobic soil volume fraction (*ansvf*), and product ratio (*pr*) [$N_2O/(N_2O+N_2)$]) (all) and a simplified model included only most important predictors ($CO_2$+*ansvf*(+*pr*); predictor (+*pr*) was only used for $N_2O$ emissions). The blue area shows additional simplified models with substitutions of the most important predictor for $O_2$ supply (*ansvf*) by $D_{sim}$ or diffusivity calculated from an empirical model ($D_{emp}$) (Deepagoda et al., 2011), or theoretical air content ($\varepsilon_t$). The red area shows a simplified model with substitutions of the most important predictor for $O_2$ demand ($CO_2$) by soil organic matter (SOM, measured in bulk soil). Substitution of both most important predictors ($CO_2$ and *ansvf*) by SOM and $D_{emp}$ is shown in the violet area.

## 4 Discussion

### 4.1 Which processes govern denitrification in soil?

The onset and magnitude of denitrification is controlled by $O_2$ supply and $O_2$ consumption, which in turn depends on processes in soil occurring at microscopic scales. This study was designed to examine different levels of $O_2$ consumptions by comparing soils with different SOM contents and different levels of $O_2$ supply by comparing different aggregate sizes and different water saturations. Other factors that would have affected $O_2$ demand and energy sources for denitrifiers (quality of organic matter, temperature, pH, plant-soil interactions), $O_2$ supply (oxygen concentration in the headspace, temperature) or other drivers of denitrification ($NO_3^-$ concentration, pH, denitrifier community structure) were either controlled or excluded in this study.

$N_2O$ release from soil can be low because denitrification does not occur under sufficient oxygen supply or because it is formed in wet soil but reduced to $N_2$ before it can escape to the atmosphere or because it is trapped in isolated air pockets (Braker and Conrad, 2011). Trapped $N_2O$ is thought to likely be reduced to $N_2$ eventually if gaseous $N_2O$ is not released after a saturation change, which would open up a continuous path to the headspace. This is shown in the schematic on the balance between $O_2$ supply and demand and its effect on denitrification (Figure 8).

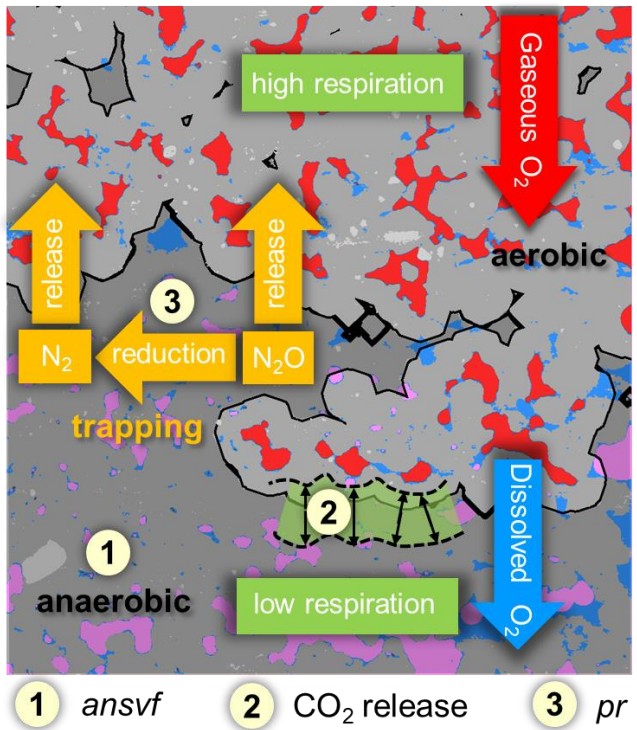

**Figure 8: Conceptual scheme of oxygen (O$_2$) supply and demand and its effect on denitrification. Material classes include soil matrix (gray area), water (blue), mineral grains (light gray), connected air (red) and isolated air (rose). The black line divides between aerobic (light gray area) and anaerobic (dark gray area) conditions. O$_2$ supply and demand regulate the formation of anaerobic soil volume fraction (*ansvf*) as an imprint of the spatial distribution of connected air (item number 1), respiration (item number 2) that would move the boundary between oxic and anoxic zones in the soil matrix closer towards the pore when soil respiration is high (and vice versa) and N$_2$O reduction to N$_2$ (expressed by the product ratio (*pr*), item number 3). The numbered items show how the explanatory variables that best describe N$_2$O release affect denitrification.**

To our knowledge, the experimental setup of the present study combined for the first time microstructure analysis of soil (X-ray CT) with measurements of N$_2$O and (N$_2$O+N$_2$) fluxes to explore controlling factors of the complete denitrification process including N$_2$ formation. The explanatory variables that contributed the highest predictive power with (N$_2$O+N$_2$) release were *ansvf* and CO$_2$ release (Figure 8). The estimated *ansvf* (item 1) is a sole function of the spatial distribution of connected air in soil and therefore only reflects soil structural properties related to O$_2$ supply. The dependence of denitrification on diffusion constraints was demonstrated by several models that were developed to predict the formation of anoxic centers within soil aggregates (Greenwood, 1961; Arah and Smith, 1989; Arah and Vinten, 1995; Kremen et al., 2005). The distance threshold for anoxic conditions to emerge was set on an ad-hoc basis at 5 mm from connected air at the end of incubation, but is likely to vary with O$_2$ demand by local microbial activity (CO$_2$ release represented by the green fringe area, item 2) during the incubation (Kremen et al., 2005; Rabot et al., 2015; Ebrahimi and Or, 2018; Keiluweit et al., 2018; Kravchenko et al., 2018; Schlüter et al., 2019). Because we could only conduct X-ray CT-scans at the end of incubation, redistribution of water during the incubation time cannot be ruled out. This could have changed *ansvf* and thus might explain some of the temporal variability of gaseous fluxes. In repacked soils it might be distributed rather uniformly and therefore correlated with bulk CO$_2$ release (Aon et al., 2001; Ryan and Law, 2005; Herbst et al., 2016). The fact that aggregate size had no effect on denitrification indicates that critical distances were larger than the aggregate radii and rather controlled by air distribution in the macropore system. When air content was high, all visible macropores where air-filled so that this critical air distance was hardly exceeded anywhere. When air content was low (close to full water saturation), the patchy distribution of air and water in the macropore system was governed by subtle layering in the pore structure and not by aggregate size. This means that both aggregate sizes used in the present study might have been too small to provoke differences in O$_2$ availability and thus in CO$_2$, N$_2$O and (N$_2$O+N$_2$) fluxes. The large

distance found here is in contrast to the very short critical distances of 180 μm for sufficient soil aeration estimated by Kravchenko et al. (2018) and Kravchenko et al. (2019) for intact soil cores containing crop residues for which soil respiration was not determined but likely to be much higher.

A somewhat surprising result is that oxygen concentration measurements did not have an added value for predicting either $N_2O$ release or total denitrification. Best correlation of local $O_2$ concentration with $\varepsilon_{con}$ was with a radial extent of 3.6 mm used for averaging around the microsensor (Figure 6). Thus, with seven microsensors per column we only probed 0.2% of the total soil volume. This is too small to capture aerobic and anaerobic conditions representatively, especially since they may switch within short distances (Figure 5). More sensors or sensors with larger support volume could be a means to improve the predictive power of local oxygen measurements. However, there is always a trade-off between retrieving more information and disturbing the soil as little as possible.

If only $N_2O$ release is concerned, *pr* as an independent proxy for $N_2O$ consumption (Figure 8 (item 3)) was beneficial to predict $N_2O$ emissions together with $CO_2$ and *ansvf* (Figure 7). The $N_2O$ reduction to $N_2$ and thus the *pr* are complexly controlled, where besides physical factors microbial (the structure of the denitrifier community) and chemical properties (pH, N oxides, SOM, temperature, salinity) are relevant (Smith et al., 2003; Clough et al., 2005; Müller and Clough, 2014). With respect to physical factors, decreasing diffusivity enhances $N_2O$ residence time and $N_2O$ concentration in the pore space thus favouring $N_2O$ reduction. According to this, Bocking and Blyth (2018) assumed a very small *pr* in wet soils, because $N_2O$ may be trapped in the soil or completely reduced to $N_2$. This assumption may also support results of the present study, where the average $(N_2O+N_2)$ fluxes peaked at the medium water saturation (particularly with GI soil) while $D_{sim}$ decreased with increasing water saturations (Figure 4), which may indicate an entrapment of $(N_2O+N_2)$ in isolated soil pores (Clough et al., 2005; Harter et al., 2016). However, $N_2$ release increased more strongly with time than the $N_2O$ release resulting in decreasing *pr* with time (Supplementary Material, Figure S5). The chance of $N_2O$ to be released before it is reduced to $N_2$ depends on the diffusion distance of dissolved (and gaseous) $N_2O$ between its formation sites and the atmosphere. Although diffusion pathways for $O_2$ and $N_2O$ are similar just in opposite direction, *ansvf* and *pr* might be a good combination of proxies to predict $N_2O$ emissions to capture physical and microbial properties.

## 4.2 How to substitute microscale information by bulk properties?

The aims of this study were to find a minimum set of variables that explain the regulation of microbial denitrification at microscopic scales in a simplified experimental setup and to explore in how far this microscopic information can be substituted by readily available bulk properties that are feasible to measure in a field campaign. The interplay of $O_2$ supply and $O_2$ demand resulted in $CO_2$ emissions and CT-derived *ansvf* being the most important predictors for $(N_2O+N_2)$ fluxes, while for $N_2O$ fluxes *pr* was also important (Figure 7, see Supplementary Material Figure S8). Simplified models with most important predictors only $(CO_2+ ansvf (+pr))$ were sufficient to achieve similar explained variabilities (81% and 83% for $N_2O$ and $(N_2O+N_2)$ fluxes, respectively) compared to the complex models. The downside of using $CO_2$ and CT-derived *ansvf* as predictors for denitrification is that these proxies are often unavailable and reasonable substitutions by easily available variables would be desirable.

The *ansvf* could have been replaced with alternative proxies for $O_2$ supply like $D_{sim}$, $D_{emp}$ and $\varepsilon_t$, which would have led to a reduction in explained variability of $(N_2O+N_2)$ fluxes to 52-78% and an even larger drop for $N_2O$ fluxes to 46-59% (Supplementary Material, Table S2). The substitution of *ansvf* by $D_{sim}$ would avoid the requirement for an ad-hoc definition of a critical pore distance threshold but it is gained with the caveat of very time-consuming 3D simulations or laborious

measurements. Therefore, the substitution of *ansvf* with diffusivity estimated by empirical models ($D_{emp}$) seems more viable. Diffusivity is mainly controlled by soil bulk density and water saturation (Balaine et al., 2013; Klefoth et al., 2014). These empirical models predict diffusivity based on empirical relationships with total porosity ($\Phi$) and air-filled porosity ($\varepsilon$) (Millington and Quirk, 1961; Moldrup et al., 2000; Resurreccion et al., 2010; Deepagoda et al., 2011; Deepagoda et al., 2019).

As expected the discrepancy between calculated $D_{emp}$ and simulated $D_{sim}$ was highest at water saturation >75% WFPS where discontinuity due to packing procedure took full effect as described earlier (Supplementary Material, Figure S9, Figure S4). The substitution of CT-derived *ansvf* by $D_{emp}$ derived from empirical models (Figure 7, Supplementary Material, Table S2) is perhaps unacceptable for a genuine understanding of $N_2O$ or ($N_2O+N_2$) emissions from individual samples since estimated diffusivity ignores the actual tortuosity and continuity of the air-filled pore space. However, it may be a promising approach to reasonably

predict average $N_2O$ or ($N_2O+N_2$) fluxes at natural conditions with readily available soil characteristics (Figure 7, Table S2). In this particular study, $D_{sim}$ could even be replaced with the theoretical air content ($\varepsilon_t$) adjusted during packing (together with $CO_2(+pr)$) without a reduction in explained variability in $N_2O$ and ($N_2O+N_2$) fluxes (Figure 7, Supplementary Material, Table S2), due to the very strong log-linear relationship between the $\varepsilon_t$ and $D_{sim}$ (Figure 4e). However, totally neglecting any proxy for $O_2$ supply, (i.e. $CO_2$ only to predict $N_2O$ fluxes), was insufficient to predict $N_2O$ fluxes (Table S2).

A different strategy to estimate *ansvf* from bulk measurements is to switch from oxic to anoxic incubation by replacing the carrier gas under otherwise constant conditions. The difference in ($N_2O+N_2$) release between the two stages will be larger, the smaller the *ansvf* during oxic incubation. Details about the calculation of this *ansvf_cal* can be found in the Supplementary Material. The *ansvf_cal* assumes that actual denitrification is linearly related to *ansvf* and that the specific anoxic denitrification rate is homogenous, i.e. would be identical at any location within the soil. Deviations from this assumption could arise from

heterogeneity in the distribution of substrates and microbial communities. However, the actual soil volume where denitrification may occur, described by the distance to aerated pores, does not only depend on $O_2$ diffusion, but also on respiration ($O_2$ consumption). Therefore, it could be expected, that *ansvf* derived from X-ray CT imaging analysis compared to *ansvf_cal* was overestimated with RM soil or underestimated with GI soil due to the differences in carbon sources and related $O_2$ consumption. The average *ansvf_cal* was similar (0.24) to the *ansvf* (0.21) for RM soil (Supplementary Material, Table S3). With GI soil,

however, the *ansvf_cal* was larger (0.45) than the image-derived *ansvf* (0.13). This difference may indeed result from an underestimation of *ansvf* of GI soil due to the higher SOM content and respiration rates. In future experiments it might be recommendable to integrate the $O_2$ consumption into *ansvf* estimation. The appeal of this two-stage incubation is that it can be conducted with larger soil columns as there is no size restriction as with the application of X-ray CT. Evidently, this two-stage incubation approach is not feasible for field campaigns, for which we would recommend to resort to estimated diffusivities

instead. However, both approaches are complementary since both are associated with different assumptions and thus uncertainties. Therefore, using them both improves the assessment of *ansvf*.

The use of $CO_2$ production as a proxy for $O_2$ demand to predict $N_2O$ and ($N_2O+N_2$) release is limited as it is not fully independent of denitrification, since anaerobic respiration contributes to total respiration. Therefore, it is appealing to replace it with estimates of microbial activity based on empirical relationships with temperature, SOM, clay and water content (Smith et

al., 2003) as these properties are routinely measured. When including the SOM measured before the experiment for the bulk soil (Table 1) to explore $N_2O$ or ($N_2O+N_2$) emissions, predictive power for ($N_2O+N_2$) decreased (63% compared to 83% with $CO_2$ instead of SOM together with *ansvf*), just like it was reduced for predicting $N_2O$ emissions (73% compared to 81% with $CO_2$ instead of SOM together with *ansvf* and *pr*). The combination of proxies for $O_2$ supply and demand, SOM and $D_{emp}$ only, to predict $N_2O$ and ($N_2O+N_2$) fluxes did not reduce the explained variability too much beyond those of individual substitutions (60

and 66%, respectively). An improvement might be achieved by accounting for different quality in SOM, e.g. mineral-associated

organic matter, fresh particulate organic matter, microbial pool; all of which will lead to different mineralisation rates and hence propensity to run into local anoxia (Beauchamp et al., 1989; Kuzyakov, 2015; Surey et al., 2020), due to the fact that SOM favours denitrification in several ways (Beauchamp et al., 1989; Ussiri and Lal, 2013), i.e. by supplying energy, leading to consume $O_2$ via respiration and supplying mineral N from mineralisation.

**4.3 Future directions and implications for modeling**

In large-scale effective N-cycling models the *ansvf* is typically linked to the partial pressure of oxygen in soil and conveys no explicit spatial information. In the long run these models like DNDC, CoupModel, MicNiT (Li et al., 1992; Jansson and Karlberg, 2011; Blagodatsky et al., 2011) might benefit tremendously from incorporating a spatially explicit *ansvf* as a state variable to predict denitrification. The estimation of *ansvf* can be improved by taking $O_2$ consumption into account. Knowledge on spatial distribution of respiration in combination with pore scale modeling would further improve *ansvf* estimations and could be used to validate our approach with oxic/anoxic incubation. However, the empirical functions to estimate this *ansvf* from readily available properties similar to empirical diffusivity models have yet to be developed and validated against a whole suite of intact soil cores with different soil types and vegetation for which oxic/anoxic incubation and X-ray CT analysis are carried out jointly.

Using intact instead of repacked soils in future experiments will represent more natural conditions, e.g. larger tortuosity and thus lower diffusivity in undisturbed compared to sieved soil (Moldrup et al., 2001). However, in undisturbed soils diffusivity and soil structure may also vary locally and as a consequence of this varying $O_2$ supply and demand affect denitrification. Under field conditions this impact on denitrification is additionally altered by saturation changes, temperature variations, atmospheric gas concentrations, microbial community structure, and plant growth. It would thus be very interesting to include also different soil types and land-use types from various climate zones in future studies, e.g. paddy soils having high water saturation and are known to show a high denitrification activity with $N_2$ emissions exceeding that of $N_2O$ emissions.

**Conclusions**

To our knowledge this is the first experimental setup combining X-ray CT derived imaging and flux measurements of complete denitrification (i.e. $N_2O$ and $(N_2O+N_2)$ fluxes) to explore the microscopic drivers of denitrification in repacked soil. We could show that changes in denitrification within different saturations could be predicted well with the anaerobic soil volume fraction (*ansvf*) estimated from image-derived soil structural properties. The differences in denitrification (i.e. $N_2O$ and $(N_2O+N_2)$ fluxes) between two investigated soils were triggered by different respiration rates due to different SOM content. A combination of CT-derived *ansvf* and $CO_2$ emission, as proxies for oxygen supply and demand, respectively, is best in predicting $(N_2O+N_2)$ emission (83% explained variability) across a large saturation range and two different soils. The product ratio (*pr*), additionally to *ansvf* and $CO_2$ emissions, was also an important predictor for emissions of only the greenhouse gas $N_2O$ (81% explained variability). The *ansvf* can also be replaced by simulated diffusivity ($D_{sim}$) (time consuming) or by diffusivity from empirical models ($D_{emp}$) but not without losing predictive power. A replacement of $CO_2$ fluxes by SOM also resulted in lower predictive power, but is recommended for large-scale applications since SOM is an independent proxy for microbial activity. The full substitution of laborious predictors (*ansvf, pr*, $CO_2$) by readily available alternatives (SOM, $D_{emp}$) reduced the explained variability to 60 and 66% for $N_2O$ and $(N_2O+N_2)$ fluxes, respectively.

The high explanatory power of image-derived *ansvf* opens up new perspectives to make predictions (e. g. by modelling approaches or in pedo-transfer functions) from independent measurements of soil structure using new techniques (e.g. X-ray CT

analysis) available today in combination with biotic properties, e. g. quantity or quality of SOM. This paves the way for explicitly accounting for changes in soil structure (e. g. tillage, plants) and climatic conditions (e. g. temperature, moisture) on
denitrification.

**Data availability.** CT data and gas emission data are available from the authors on request.

**Author contribution.** H-JV, RW and SS designed the experiment. SS, BA and LR carried out the experiment. G-MW developed the statistical analysis. SS and LR prepared the manuscript with contributions from all co-authors.

*Competing interests.* The authors declare that they have no conflict of interest.

*Acknowledgments.* We thank Jürgen Böttcher from the Institute of Soil Science, Leibniz University in Hannover, for measurements of soil materials used for incubation and Anette Giesemann and Martina Heuer from Thünen-Institute for Climate-Smart Agriculture in Braunschweig, Germany, for isotopic analysis. Our thanks go to Ines Backwinkel und Jan-Reent Köster from Thünen-Institute for Climate-Smart Agriculture in Braunschweig, Germany, for conducting parallel incubations under oxic and anoxic conditions. This study is funded by the Deutsche Forschungsgemeinschaft through the research unit research unit
DFG-FOR 2337: Denitrification in Agricultural Soils: Integrated Control and Modelling at Various Scales (DASIM), grant umber 270261188.

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
