# Peer review of "Denitrification in soil as a function of oxygen availability at the microscale"

_Biogeosciences, 2020_

## Author Comment (AC1) · 20 Aug 2020

We have discovered that we unfortunately made an error in the calculation of nitrogen (N) loss during the incubation of repacked soil cores (see results section 3: 3.1 Bulk respiration). Instead of the N loss values given in the manuscript (2-4% at $\geq$75% WFPS with RM soil or 5-14% at $\geq$75% WFPS with GI soil), the corrected values of N loss are much smaller, $\leq$0.12% with RM soil and $\leq$0.38% with GI soil. We will include the corrected values of N loss in the revised manuscript, but would like to take the opportunity to report about the incorrect values in the current version. We would like to point out that the correction of N loss values has no effect on calculated fluxes or the validity of interpretations of the results.

---

## Referee Comment (RC1) · Anonymous Referee #1 · 24 Aug 2020

This manuscript investigated the effects of aggregate size and water saturation on N2O and N2 fluxes in two soils with contrasting SOM content by repacked soil cores based 15N tracer incubation in combination with X-Ray computed tomography. The main outcome was that N-gases emissions could be well predicted by considering proxies for oxygen supply (anaerobic soil volume fraction, i.e., ansvf) and demand (CO2 emissions), which linked the change of soil structure with N-gases emissions. Generally, this manuscript is well prepared and written, and the conclusions were supported by the results of the experiments. One of my major concerns was that how could one time point (at the end of the incubation) microstructure analysis for the repacked soil cores represent the change of ansvf during the 192 h lasting incubation. In addition, why the aggregate size exhibit no obvious effects on CO2 and denitrification product

stoichiometry should be discussed. Specific comments Introduction The challenge for direct measuring soil borne N2 from soil cores should be mentioned. This info may also provide rational for the authors to use 15N tracer to estimate N2 flux. Results I suggest move the resulting regression equations from SI to text so that the reader could easily capture the key point of explanatory variables for denitrification. Line 23,567 oxygen should be O2 Line 24, I suggest change the order of "ansvf" and "CO2" since "CO2" is more important in terms of explanatory based on the author's results. Line 119, comma in the sentence should be deleted. Line 151, why additional nitrate solution was sprayed in the last two treatments? if the N substrates differed among the three treatments, how could the author compared the N2O and N2 flux among the tree treatments? Line 222, clearly Line 444-445 the order of the sub figures for the two tested soils was reversed Line 545 is?

---

## Author Comment (AC2) · 27 Aug 2020

We thank the reviewer for the constructive comments. Please refer to the attached pdf file that contains answers to each comment.

Please also note the supplement to this comment:
https://bg.copernicus.org/preprints/bg-2020-221/bg-2020-221-AC2-supplement.pdf

---

## Referee Comment (RC2) · Anonymous Referee #2 · 7 Sep 2020

Denitrification process is of critical importance because it is closely related with agricultural sustainability, environmental quality, and human health. However, the denitrification process in particular N2O/N2 generation and emission is poorly understood at microscopic scales. This study provides very useful information towards understanding the complete denitrification process with X-ray CT imaging analysis, and gives new insights how the N2O and N2O+N2 are formed in soils at microscopic scales.

Major issues/concerns

The authors selected two different land use types of soils when investigating soil organic matter contents. The grassland soil has a SOM up to 4.5%, much higher than that of arable soil. I feel that it is difficult to compare the denitrification process between soils with different land use types. The authors had better use arable soils with different gradients of SOM to investigate the effects of SOM on the complete denitrification process.

It is unclear why the authors set up these three different water saturation (70, 83 and 95%). The 60% water saturation is widely used when setting up the soil microcosm experiments. I feel that 60% water saturation is needed as the control when setting up the gradients of water saturation experiment in this study. Moreover, the flooded paddy soils are widely distributed all over the world, in particular Asian areas. The authors had better include such kind of soil in their experiments to gain a full picture of water saturation effects on the complete denitrification process. The flooded paddy soil usually has a low N2O emission but a high N2 emission. It may be an excellent material when investigating the effects of water saturation on the complete denitrification.

The authors have shown very detailed information in Results section. However, it is difficult for reader to follow in this section. So the authors need to improve this section and lead the readers to pay attention to their important findings.

The authors showed their results based on different gradients of distance, water saturation and so on. I feel that they need to show their results with incubation time, at least in supplementary files. They should clarify why they show the results of a specific incubation time in the main body of this manuscript.

Minor issues/concerns

P4 L119: delete the comma after N2O.

P5 L150-151: is added nitrate amounts equal for each treatment?

P24 L630-633: please clarify this sentence.

---

## Referee Comment (RC3) · Anonymous Referee #3 · 8 Sep 2020

This study aimed to explore the controlling factors (soil organic matter, aggregate size, water saturation) of the denitrification process (N2O/N2) at microscopic scale using new approaches of X-ray computed tomography and 15N tracer incubation. They found that N2O/N2 fluxes could be well predicted by anaerobic soil volume fraction (ansvf, O2 supply) and CO2 release (O2 demand). This findings would expand our understanding of how the N2O and N2 are formed in soils. In general, the experimental design is clear, and the manuscript is well written. However, there are some concerns about the methodology and data interpretation. Major comments 1. The authors selected two types of soils with contrasting soil properties, including soil organic matter contents, soil texture, soil pH and etc., so it is unclear why the authors concluded the differences in denitrification (N2O and N2O+N2 fluxes) between two investigated soils were

triggered by different respiration rates due to different SOM content rather than other properties. 2. In the results section, the authors displayed the averages for the whole incubation, I feel it is better to show their results with incubation time. And of course, I also think it is not so reasonable to correlated average gas fluxes to the X-ray CT data of a specific incubation time, because the fluxes are not constant during incubation, neither for anaerobic soil volume fraction. 3. From the detailed information showed in supplementary file, the variation between three replicates is very large (eg. Figure S1), the reasons for this large variation as well as the effects on the data reliability need to be clarify. 4. And of course it would have been of interest to see the variations in denitrifying communities at microscopic scale. Minor comments: L125: The soil depth of topsoil should be define, 0-20 cm? L141: How much soil is used for each column? How about the soil depth of the repacked soil cores? How to control the compactness of filling? L150-151: Why spray additional nitrate solution in 83% and 95% WHC treatments but not in 70% WHC?

---

## Author Comment (AC3) · 23 Oct 2020

***Additional comment on***
***Review response on*** **"Denitrification in soil as a function of oxygen supply and demand at the microscale"** ***by*** **Lena Rohe et al.**
**Anonymous Referee #1**

*Reconsidering our data in detail revealed a mistake in calculating the fluxes of $CO_2$, $N_2O$ and $(N_2O+N_2)$. This error occurred because of wrong parentheses in the calculation. Correcting the calculation revealed increased values of fluxes by a constant factor compared to the previous values. All calculated fluxes have been corrected, having effects on $CO_2$, $N_2O$ and $(N_2O+N_2)$ fluxes, N loss and Figure 3, Figure 5 (will be removed to Supplementary Material), Figure S1, S3, Table S1 and S4, and the explained variability of $N_2O$ and $(N_2O+N_2)$ fluxes (calculated by the partial least square regression; PLSR) (Figure 8, Figure S7 and Table S2). We want to point out, that the values of fluxes are higher in the revised version, although the course of $CO_2$, $N_2O$ and $(N_2O+N_2)$ fluxes over incubation time did not change. We apologize very much for this mistake, but the changes made because of the increased fluxes did not affect the interpretation of data or statements of our study.*

*In the meantime we were able to calculate the ansvf ($ansvf_{cal}$) from parallel incubations using $(N_2O+N_2)$ fluxes during oxic conditions and after switching to anoxic conditions (Supplementary Material). Therefore, instead of reporting $ansvf_{cal}$ based on the comparison between oxic and anoxic $(N_2O+N_2)$ fluxes of two different incubation experiments, we now report values based on fluxes of the same experiment which we consider more reliable. Although $ansvf_{cal}$ values changed slightly our previous conclusions remain unchanged.*

---

## Author Response (AR3)

**Point-by-point responses on reviews on* **"Denitrification in soil as a function of oxygen supply and demand at the microscale"** *by* **Lena Rohe et al.**

*We thank the editor and three reviewers very much for the positive opinion and constructive comments on the manuscript.*
*The authors' answer is in italic font.*

*Reconsidering our data in detail revealed a mistake in calculating the fluxes of $CO_2$, $N_2O$ and $(N_2O+N_2)$. This error occurred because of wrong parentheses in the calculation. Correcting the calculation revealed increased values of fluxes by a constant factor compared to the previous values. All calculated fluxes have been corrected, having effects on $CO_2$, $N_2O$ and $(N_2O+N_2)$ fluxes, N loss and Figure 3, Figure 5 (was removed to Supplementary Material, Figure S6), Figure S1, S3, Table S1 and S4, and the explained variability of $N_2O$ and $(N_2O+N_2)$ fluxes (calculated by the partial least square regression; PLSR) (Figure 7 in revised version), Figure S8 in revised version and Table S2). We want to point out, that the values of fluxes are higher in the revised version, although the course of $CO_2$, $N_2O$ and $(N_2O+N_2)$ fluxes over incubation time did not change. We apologize very much for this mistake, but the changes made because of the increased fluxes did not affect the interpretation of data or statements of our study.*

*In the meantime we were able to calculate the ansvf ($ansvf_{cal}$) from parallel incubations using $(N_2O+N_2)$ fluxes during oxic conditions and after switching to anoxic conditions (Supplementary Material, Table S3). Therefore, instead of reporting $ansvf_{cal}$ based on the comparison between oxic and anoxic $(N_2O+N_2)$ fluxes of two different incubation experiments, we now report values based on fluxes of the same experiment which we consider more reliable. Although $ansvf_{cal}$ values changed slightly our previous conclusions remain unchanged.*

**Anonymous Referee #1**
Referee: This manuscript investigated the effects of aggregate size and water saturation on $N_2O$ and $N_2$ fluxes in two soils with contrasting SOM content by repacked soil cores based [15]N tracer incubation in combination with X-Ray computed tomography. The main outcome was that N-gases emissions could be well predicted by considering proxies for oxygen supply (anaerobic soil volume fraction, i.e., ansvf) and demand ($CO_2$ emissions), which linked the change of soil structure with N-gases emissions. Generally, this manuscript is well prepared and written, and the conclusions were supported by the results of the experiments.

One of my major concerns was that how could one time point (at the end of the incubation) microstructure analysis for the repacked soil cores represent the change of ansvf during the 192 h lasting incubation.

*In theory the anaerobic soil volume fraction (ansvf) should be governed by $O_2$ supply imprinted by the distribution of air-filled pores and modulated locally by the $O_2$ demand through microbial respiration. The former was estimated from CT derived images after 192 h of incubation using the distance to air filled pores as an estimate caused only by physical conditions, i. e. pore structure (connected air), as explained in the method section (line 224 ff.).*

*The reviewer is correct in criticizing that we cannot rule out redistribution of water and air during 192 h of incubation. We assume that such redistribution events are typically associated with abrupt changes in local $O_2$ concentrations as well as $CO_2$ and $N_2O$ release. The time series data (Figures S1 and S2) show that this may occur occasionally. However, taking several CT scans during incubation was just not an option due to methodological challenges. Likewise, variations of ansvf due to $O_2$ demand by local microorganisms (i.e. activity) and over incubation time cannot be estimated. However, in the discussion section variations of ansvf due to $O_2$ demand were mentioned (line 496 ff. and line 596 ff.).*

*We assume that there are substantial variation during the first 24 h of incubation, which are omitted from the analysis, but only minor variations after all the genes for denitrification have been expressed and the soil has reached a dynamic equilibrium of $O_2$ supply and demand and a rather static distribution of water and air. Although microbial activity could affect the ansvf, ansvf largely contributed to explanation of $N_2O$ and $(N_2O+N_2)$ fluxes, in combination with $CO_2$ release.*

*Another method was also used to estimate the ansvf (ansvf$_{cal}$, see Supplementary Material) by microbial denitrification activity only. We found accordance between both estimates for RM soil and discussed possible reasons for differences between ansvf and ansvf$_{cal}$ for GI soil.*

*In the revised version we discussed in more detail that ansvf may be altered by $O_2$ demand ($CO_2$ release) and/or $O_2$ supply during the incubation time of 192 h (l. 496 ff): "The distance threshold for anoxic conditions to emerge was set on an ad-hoc basis at 5 mm from connected air at the end of incubation, but is likely to vary with $O_2$ demand by local microbial activity ($CO_2$ release represented by the green fringe area, item 2) during the incubation (Kremen et al., 2005; Rabot et al., 2015; Ebrahimi and Or, 2018; Keiluweit et al., 2018; Kravchenko et al., 2018; Schlüter et al., 2019). Because we could only conduct X-ray CT-scans at the end of incubation, redistribution of water during the incubation time cannot be ruled out. This could have changed ansvf and thus might explain some of the temporal variability of gaseous fluxes."*

Referee: In addition, why the aggregate size exhibit no obvious effects on CO2 and denitrification product stoichiometry should be discussed.

*In the present study aggregate size did not affect $CO_2$ release or denitrification and we argued that aggregate radii (1-2 or 2-4 mm) were smaller than the thresholds of distances to connected air that were found to determine the ansvf. The critical distance to estimate the ansvf were selected from best correlations between ansvf and $N_2O$ as well as $(N_2O+N_2)$ fluxes. Results indicated that aggregate size might have been too small to provoke differences in $CO_2$, $N_2O$ and $(N_2O+N_2)$ fluxes. This point will be considered in more detail in the revised version.*

*So far we discussed this point in line 503 ff.: "The fact that aggregate size had no effect on denitrification indicates that critical distances were larger than the aggregate radii and rather controlled by air distribution in the macropore system. This is in contrast to the very short critical distances of 180µm for sufficient soil aeration estimated by Kravchenko et al. (2018) and Kravchenko et al. (2019) for intact soil cores containing crop residues for which soil respiration was not determined but likely to be much higher."*

Referee: Specific comments Introduction
The challenge for direct measuring soil borne $N_2$ from soil cores should be mentioned. This info may also provide rational for the authors to use [15]N tracer to estimate $N_2$ flux.

*We agree that this point could be better introduced and was rephrased in the updated manuscript as (line 85 ff.):*

*"Since the $N_2$ background of air (78%) is very high, direct $N_2$ measurement from denitrification in soil is very challenging (Groffman et al., 2006; Mathieu et al., 2006). The $^{15}N$ labelling technique is a method successfully applied to determine $N_2O$ and also $N_2$ production from denitrification from $^{15}N$ amended electron acceptors ($NO_3^-$) (Mathieu et al., 2006; Scheer et al., 2020)." Moreover, we explained that $N_2$ depleted atmosphere was used to improve $N_2$ flux detection (l. 82 ff.).*

Referee: Results
I suggest move the resulting regression equations from SI to text so that the reader could easily capture the key point of explanatory variables for denitrification.
*This is a good remark and we moved the regression equations to the main text in the revised version (Result section, 3.4 Explanatory variables for denitrification, l. 442 ff.).*

Referee: Line 23,567 oxygen should be O2
*We replaced oxygen with $O_2$ in the revised version.*

Referee: Line 24, I suggest change the order of "ansvf" and "CO2" since "CO2" is more important in terms of explanatory based on the author's results.
*We changed the order of $CO_2$ and ansvf in the revised version.*

Referee: Line 119, comma in the sentence should be deleted.
*We deleted the comma in the revised version.*

Referee: Line 151, why additional nitrate solution was sprayed in the last two treatments? if the N substrates differed among the three treatments, how could the author compared the N2O and N2 flux among the tree treatments?
*We agree that this should be clarified and explained in more detail. All treatments contained the same amount of nitrate per mass of soil (50mg/kg soil). Hence the total amount of nitrate per column differed between the two soil types due to different bulk densities. However, the total amount of nitrate did not differ between three saturation levels. 50mg/kg $N$-$KNO_3$ was added to the respective amount of water. Hence, for higher water saturations the nitrate concentration in the solution was lower, so that the total amount was the same. This solution was used for moistening the soil. We rephrased as (l. 132 ff.):*
*"Three different saturation treatments were prepared for subsequent incubation experiments (70%, 83% and 95% WHC) to control the $O_2$ supply and thus provoke differences in denitrification activity. A $^{15}N$ solution was prepared by mixing 99 at% $^{15}N$-$KNO_3$ (Cambridge Isotope Laboratories, Inc., Andover, MA, USA) and unlabelled $KNO_3$ (Merck, Darmstadt, Germany) to reach 50 mg N $kg^{-1}$ soil with 60 at% $^{15}N$-$KNO_3$ in each water saturation treatment. Hence, for higher water saturations the stock solution was more diluted in order to reach the same target concentration in the soil. In a first step the soil was adjusted to 70% WHC before packing. [...] For the latter two saturation levels the rest of $NO_3^-$ solution was sprayed sequentially onto each layer after packing."*

Referee: Line 222, clearly
*We replaced "clearaly" by "clearly" in the revised version.*
*"Only macropores twice this nominal resolution were clearly detectable in the soil core images."*

Referee: clearly Line 444-445 the order of the sub figures for the two tested soils was reversed

*We corrected this mistake:*

*"Figure 7: Average O₂ saturation (at the end of incubation experiment) measured with 4 sensors each located at the center of soil core as a function of distance to visible connected air for soil from Gießen (GI, (a)-(c), blue) and Rotthalmünster (RM, (d)-(f), red), and for two aggregate sizes (2-4mm and 4-8mm). (a) and (d) show results for lowest (b) and (e) for medium and (c) and (f) for highest water saturation. The inset in (a), (b), and (d) shows a reduced distance range. The distance to visible connected air is averaged in a spherical region around the sensor tip (7.2 mm diameter). The Spearman's rank correlation coefficient (R) result from Spearman's rank correlation and indicate the extent of monotonic relation between the ranks of both variables. The associated p-values (p) were corrected for multiple comparison according to Benjamini and Hochberg (1995)."*

Referee: Line 545 is?

> *We wrote "as" instead of "is".*
>
> *"However, there is always a trade-off between retrieving more information and disturbing the soil as little as possible."*

**Anonymous Referee #2**

Denitrification process is of critical importance because it is closely related with agricultural sustainability, environmental quality, and human health. However, the denitrification process in particular $N_2O/N_2$ generation and emission is poorly understood at microscopic scales. This study provides very useful information towards understanding the complete denitrification process with X-ray CT imaging analysis, and gives new insights how the $N_2O$ and $N_2O+N_2$ are formed in soils at microscopic scales.

Major issues/concerns

The authors selected two different land use types of soils when investigating soil organic matter contents. The grassland soil has a SOM up to 4.5%, much higher than that of arable soil. I feel that it is difficult to compare the denitrification process between soils with different land use types. The authors had better use arable soils with different gradients of SOM to investigate the effects of SOM on the complete denitrification process.

> *We acknowledge that grassland and agricultural soil have vastly different soil structure and different input of plant residues. However, these effects are removed after sieving and removal of particulate organic matter and long-term storage. In other words, we did not work with differently managed soil, but rather with soil material with similar texture, but different SOM content, artificially repacked to some target bulk density, so that potential management effects are ruled out.*
>
> *In our experiment we controlled the nitrate content, temperature and water saturation, but could include other measures for oxygen supply and demand, such as soil structure measures that are influenced by the soil texture (i. e. proportion of sand, silt and clay in soil), or $CO_2$ fluxes that indicate microbial activity. Possibilities to explore complete denitrification with soil organic matter (SOM) were described in detail in the discussion section (l. 576 ff.).*
>
> *However, experiments including variations in temperature, nitrate availability or other properties, like SOM gradients would be very interesting and expand the knowledge on denitrification.*

It is unclear why the authors set up these three different water saturation (70, 83 and95%). The 60% water saturation is widely used when setting up the soil microcosm experiments. I feel that 60% water saturation is needed as the control when setting up the gradients of water saturation experiment in this study.

> *It is true, a lower water saturation is widely used, especially in studies focussing on nitrification or on co-occuring processes like nitrification, nitrifier denitrification and denitrification. It is known from previous studies, that $N_2O$ is produced during nitrification in*

*soil at approximately 70% WFPS (Davidson 1991, Cardenas et al. 2017). This paper focuses on denitrification only. So with a series from 63% to 95% WFPS we capture the transition from low $N_2O$ production through denitrification due to sufficient oxygen supply all the way to low $N_2O$ emission due to further reduction to $N_2$. Another treatment would not have brought about any additional insights into the microscale mechanisms at play. Moreover, we conducted pre-test with varying WFPS, finding that with these soils, minimum saturation of 75% WFPS was necessary to ensure robust $N_2$ flux detection.*

Moreover, the flooded paddy soils are widely distributed all over the world, in particular Asian areas. The authors had better include such kind of soil in their experiments to gain a full picture of water saturation effects on the complete denitrification process. The flooded paddy soil usually has a low $N_2O$ emission but a high $N_2$ emission. It may be an excellent material when investigating the effects of water saturation on the complete denitrification.

*It is true, that water saturation effects on complete denitrification of paddy soils, in particular differences in $N_2O$ and $N_2$ emissions following different saturations, is very interesting to analyse, especially when regarding effects of climate change on such anthropogenic systems. However, naturally these flooded or ponded tropical or subtropical soils are exposed to completely different climatic conditions than the selected soils of the presented study. Thus it might be very interesting to include such soils in comparable experiments with temperature gradients as an additional factor for denitrification activity. The current study focussed on disentangling structural effects of mineral soils on $O_2$ supply and $O_2$ demand, without considering of temperature effects.*

*We have touched this comment in the section 4.3. (Future directions and implications for modelling) and included in the updated version at the end of this section (l. 606 ff.):*

*"It would thus be very interesting to include also different soil types and land-use types from various climate zones in future studies, e.g. paddy soils having high water saturation and are known to show a high denitrification activity with $N_2$ emissions exceeding that of $N_2O$ emissions.*

The authors have shown very detailed information in Results section. However, it is difficult for reader to follow in this section. So the authors need to improve this section and lead the readers to pay attention to their important findings.

*Thank you for the suggestion. We tried to sharpen the results section by removing the regression analysis of ansvf with different gases into the supporting information and only keeping the essential findings of this regression analysis in the main text. By this, we have removed one figure (Figure 5) and one paragraph from the main paper.*

The authors showed their results based on different gradients of distance, water saturation and so on. I feel that they need to show their results with incubation time, at least in supplementary files. They should clarify why they show the results of a specific incubation time in the main body of this manuscript.

*Structural measures were only analysed at the end of incubation. $CO_2$ and $N_2O$ fluxes, $O_2$ consumption, and product ratios are presented as a function of time in the Supplementary Material (Figure S1, S2 and S5). Average values of $CO_2$, $N_2O$ and ($N_2O+N_2$) release of the incubation period (24-192 h) were used for correlations. Average $O_2$ saturation of the final 24 h was taken for all subsequent analysis, as this probably best reflects the water distribution scanned with X-ray CT (see l. 315 ff.).*

*Regarding the CT derived measures (e. g. connected air, diffusivity, distance to connected air, ansvf), the reviewer is correct in criticizing that we cannot rule out redistribution of water and air during 192 h of incubation. We assume that such redistribution events are typically associated with abrupt changes in local $O_2$ concentrations as well as $CO_2$ and $N_2O$ release. The time series data (Figures S1 and S2) show that this may occur occasionally. However,*

*taking several CT scans during incubation was just not an option due to methodological challenges. Likewise, variations of ansvf due to $O_2$ demand by local microorganisms (i.e. activity) and over incubation time cannot be estimated. We assume that there are substantial variation during the first 24 h of incubation, which are omitted from the analysis, but only minor variations after all the genes for denitrification have been expressed and the soil has reached a dynamic equilibrium of $O_2$ supply and demand and a rather static distribution of water and air. Although microbial activity could affect the ansvf, ansvf largely contributed to explanation of average $N_2O$ and ($N_2O+N_2$) fluxes, in combination with $CO_2$ release.*

Minor issues/concerns

P4 L119: delete the comma after N2O.

*We deleted the comma in the revised version.*

P5 L150-151: is added nitrate amounts equal for each treatment?

*We agree that this should be clarified and explained in more detail. All treatments contained the same amount of nitrate per mass of soil (50mg/kg soil). Hence the total amount of nitrate per column differed between the two soil types due to different bulk densities. However, the total amount of nitrate did not differ between three saturation levels. 50mg/kg N-KNO₃ was added to the respective amount of water. Hence, for higher water saturations the nitrate concentration in the solution was lower, so that the total amount was the same. This solution was used for moistening the soil. We rephrased as (l. 132 ff.):*

*"Three different saturation treatments were prepared for subsequent incubation experiments (70%, 83% and 95% WHC) to control the $O_2$ supply and thus provoke differences in denitrification activity. A $^{15}N$ solution was prepared by mixing 99 at% $^{15}N$-KNO₃ (Cambridge Isotope Laboratories, Inc., Andover, MA, USA) and unlabelled KNO₃ (Merck, Darmstadt, Germany) to reach 50 mg N kg$^{-1}$ soil with 60 at% $^{15}N$-KNO₃ in each water saturation treatment. Hence, for the two higher water saturations the stock solution was more diluted in order to reach the same target concentration in the soil. In a first step the soil was adjusted to 70% WHC before packing. [...]For the latter two saturation levels the rest of NO₃$^-$ solution was sprayed sequentially onto each layer after packing."*

P24 L630-633: please clarify this sentence.

*This sentence was removed without loss.*

**Anonymous Referee #3**

This study aimed to explore the controlling factors (soil organic matter, aggregate size, water saturation) of the denitrification process (N2O/N2) at microscopic scale using new approaches of X-ray computed tomography and 15N tracer incubation. They found that N2O/N2 fluxes could be well predicted by anaerobic soil volume fraction (ansvf, O2 supply) and CO2 release (O2 demand). This findings would expand our understanding of how the N2O and N2 are formed in soils. In general, the experimental design is clear, and the manuscript is well written. However, there are some concerns about the methodology and data interpretation.

Major comments

1.The authors selected two types of soils with contrasting soil properties, including soil organic matter contents, soil texture, soil pH and etc., so it is unclear why the authors concluded the differences in denitrification (N2O and N2O+N2 fluxes) between two investigated soils were triggered by different respiration rates due to different SOM content rather than other properties.

*Main drivers for soil respiration are temperature, water saturation, oxygen saturation and nutrient content / availability. Soil types in turn affect soil structure, i. e. water saturation and oxygen saturation, and nutrient availability. The temperature was set at 20 °C during the*

*incubation experiment and the water saturation was controlled in parallel experiments (70, 83, 90 % WHC). It is true, that soil texture or soil pH might affect the nutrient storage and thus availability for microbes, but nitrate as the electron acceptor for denitrification was supplied sufficiently in the presented experiment. Thus we could exclude the availability of nitrate, temperature effects, or water saturation in our analysis. In the revised version we included, that a recent study by Malique et al. (2019) investigated the denitrification potential of both soils (RM and GI) and found a higher denitrification potential with GI soil compared to that of RM soil. This finding emphasizes that soil texture and bulk density should mainly govern air content and thus $O_2$ supply at a certain water saturation, whereas SOM content should mainly govern microbial activity and thus $O_2$ demand and energy sources for denitrifiers. We fully account for bulk density differences through its effect on air content and air distribution at a given water saturation. This is assessed by proxies for $O_2$ supply.*

*We described controlled or excluded factors at the beginning of the discussion as follows (l. 471 ff.): "This study was designed to examine different levels of $O_2$ consumptions by comparing soils with different SOM contents and different levels of $O_2$ supply by comparing different aggregate sizes and different water saturations. Other factors that would have affected $O_2$ demand and energy sources for denitrifiers (quality of organic matter, temperature, pH, plant-soil interactions), $O_2$ supply (oxygen concentration in the headspace, temperature) or other drivers of denitrification ($NO_3^-$ concentration, pH, denitrifier community structure) were either controlled or excluded in this study. "*

*However, experiments including variations in temperature, nitrate availability and/or other properties, like SOM or soil structure, would be very interesting and expand the knowledge on denitrification.*

2. In the results section, the authors displayed the averages for the whole incubation, I feel it is better to show their results with incubation time. And of course, I also think it is not so reasonable to correlated average gas fluxes to the X-ray CT data of a specific incubation time, because the fluxes are not constant during incubation, neither for anaerobic soil volume fraction.

*Structural measures were only analysed at the end of incubation. $CO_2$ and $N_2O$ fluxes, $O_2$ consumption, and product ratios are presented as a function of time in the Supplementary Material (Figure S1, S2 and S5). Average values of $CO_2$, $N_2O$ and ($N_2O+N_2$) release of the incubation period (24-192h) were used for correlations. Average $O_2$ saturation of the final 24h was taken for all subsequent analysis, as this probably best reflects the water distribution scanned with X-ray CT (see line 315 ff.).*

*Regarding the CT derived measures (e. g. connected air, diffusivity, distance to connected air, ansvf), the reviewer is correct in criticizing that we cannot rule out redistribution of water and air during 192 h of incubation. We assume that such redistribution events are typically associated with abrupt changes in local $O_2$ concentrations as well as $CO_2$ and $N_2O$ release. The time series data (Figures S1 and S2) show that this may occur occasionally. However, taking several CT scans during incubation was just not an option due to methodological challenges. Likewise, variations of ansvf due to $O_2$ demand by local microorganisms (i.e. activity) and over incubation time cannot be estimated (line 496 ff.).*

*We assume that there are substantial variation during the first 24 h of incubation, which are omitted from the analysis, but only minor variations after all the genes for denitrification have been expressed and the soil has reached a dynamic equilibrium of $O_2$ supply and demand and a rather static distribution of water and air. Although microbial activity could affect the ansvf, ansvf largely contributed to explanation of $N_2O$ and ($N_2O+N_2$) fluxes, in combination with $CO_2$ release.*

3. From the detailed information showed in supplementary file, the variation between three replicates is very large (eg. Figure S1), the reasons for this large variation as well as the effects on the data reliability need to be clarify.

*We can only assume possible reasons for the observed variations between replicates, but since the replicates were treated very similar according to the described protocol we cannot clearly identify reasons. The only explanation that we found was that small differences in repacking the moistened soil aggregates occurred between replicates (i. e. compaction, distribution of pores, and connectivity of pores), and possibly heterogeneity in the content of organic matter fractions in the aggregates. These small differences may largely affect soil aeration und thus microbial activity.*

*As can be clearly seen in Figure S4, repacking the aggregates in 2 cm intervals affected the visible air content and also ansvf. Both measures largely differed among replicates incubated at medium saturation for GI and RM soil. This was also pointed out in the result section 3.2, l 350 ff..*

*For $CO_2$ emission it was discussed in line 300 ff.: "The variability in $CO_2$ fluxes between replicates is much higher than the temporal variability during incubation. This is probably explained by small differences in packing of the columns that can have large consequences for soil aeration."*

*The same explanation was given for $N_2O$ and $(N_2O+N_2)$ emissions in line 305 ff.: "The huge variability between replicates is again higher than the temporal variability (e.g. in Figure 4d and time series in Supplementary Material, Figure S1) and the effect of aggregate size is inconsistent due to the large variability among replicates." Additionally, small variations in $N_2O$ emissions may result from co-occuring $N_2$ emissions during denitrification.*

*Regarding the $O_2$ saturation averages of the last 24h of incubation were used for correlations and statistical analysis, because we assumed best accordance of the $O_2$ averages and water distribution (connected air content and ansvf) analysed by CT image analysis at the end of the experiment. The reliability of $O_2$ saturation data was discussed in the results section (l. 316 ff.): "Average $O_2$ saturation was lowest with highest water saturation and roughly the same for saturations <80%WFPS (Figure 3b). Some sensors showed a gradual decline in $O_2$ concentration, whereas some showed a drastic reduction or increase in a short period of time, probably due to water redistribution (Supplementary Material, Figure S2). The average of the final 24h was taken for all subsequent analysis, as this probably best reflects the water distribution scanned with X-ray CT. Standard errors among the seven $O_2$ microsensors were high in each treatment due to very local measurement of $O_2$ that probed very different locations in the heterogeneous pore structure."*

*We think that the data are reliable and comparable, because $CO_2$, $N_2O$ and $(N_2O+N_2)$ emissions and $O_2$ saturation as well as the other explanatory variables of the present study were measured for each replicate. Thus, small variations in connected air content or ansvf affect denitrification and respiration in one soil core.*

4. And of course it would have been of interest to see the variations in denitrifying communities at microscopic scale.

*We agree that this information would be very interesting and helpful for interpretation of results. However, we have presented a very comprehensive experimental setup, combining gas flux measurements, isotopic analysis, image analysis of CT derived data as well as simulating the diffusivity. These were very time consuming methods, especially the demanding image analysis. Methods to analyse the denitrifying communities in soil are not established in our lab and unfortunately we were not able to perform genetic analysis. In the revised version the microbial community was added to the other factors altering denitrification under field conditions in the section 4.3 "Future directions and implications for modelling" (l. 606 ff.).*

Minor comments:

L125: The soil depth of topsoil should be define, 0-20 cm?

*This information was included as follows (l. 117 ff.): "Fine-textured topsoil material was collected from two different agricultural sites in Germany (from a depth of 10 - 20 cm in Rotthalmünster (RM) and of 3 - 15 cm in Gießen (GI) as representatives for agricultural mid-European soils (Table 1).*

L141: How much soil is used for each column?

*The target bulk density was 1.3 g cm$^{-3}$ for RM soil and 1.0 g cm$^{-3}$ for GI soil (Table 1). Thus 902 g dry weight of RM soil and 694 g dry weight of GI soil were used per column.*

*In line 145, we included: "This packing resulted in 902 and 694 g dry weight of RM and GI soil, respectively."*

How about the soil depth of the repacked soil cores?

*The height of the repacked soil cores was 10 cm. This information is provided in line 141 and Figure 1.*

How to control the compactness of filling?

*We repacked the soil in five 2 cm intervals (l. 141 ff.).*

*"This $^{15}$N-labelled soil was filled in 2 cm intervals into cylindrical PVC columns (9.4cm inner diameter x10cm height) (Figure 1) and compacted to a target bulk density that correspond to site-specific topsoil bulk densities (Jäger et al., 2003; John et al., 2005)."*

L150-151: Why spray additional nitrate solution in 83% and 95% WHC treatments but not in 70% WHC?

*We agree that this should be clarified and explained in more detail. All treatments contained the same amount of nitrate per mass of soil (50mg/kg soil). Hence the total amount of nitrate per column differed between the two soil types due to different bulk densities. However, the total amount of nitrate did not differ between three saturation levels. 50mg/kg N-KNO$_3$ was added to the respective amount of water. Hence, for higher water saturations the nitrate concentration in the solution was lower, so that the total amount was the same. This solution was used for moistening the soil. We rephrased as (l. 132 ff.):*

*"Three different saturation treatments were prepared for subsequent incubation experiments (70%, 83% and 95% WHC) to control the O$_2$ supply and thus provoke differences in denitrification activity. A $^{15}$N solution was prepared by mixing 99 at% $^{15}$N-KNO$_3$ (Cambridge Isotope Laboratories, Inc., Andover, MA, USA) and unlabelled KNO$_3$ (Merck, Darmstadt, Germany) to reach 50 mg N kg$^{-1}$ soil with 60 at% $^{15}$N-KNO$_3$ in each water saturation treatment. Hence, for the two higher water saturations the stock solution was more diluted in order to reach the same target concentration in the soil. In a first step the soil was adjusted to 70% WHC before packing. [...] Packing in five vertical intervals achieved a uniform porosity across the column. However, there were inevitable porosity gradients within intervals (Figure S4) that affected the air and water distribution and thus air continuity at high water saturations. This packing resulted in 902 and 694 g dry weight of RM and GI soil, respectively. For the latter two saturation levels the rest of NO$_3^-$ solution was sprayed sequentially onto each layer after packing."*

**The editor has some minor concerns**

(1) Pls briefly why upland soil is selected because grassland and arable soils are often exposed to the atmosphere containing approximately 20% oxygen. Therefore, it seems that denitrifiers may be favored only in the microanaerobic site. This kind of upland soil may be dominated by aerobic process, particularly for the surface soil.

*The soils were selected as representatives for agricultural mid-European soils. Previous studies with topsoil from Gießen could already prove its denitrification potential (e. g. Müller et al. (2002); Müller et al. (2014)). Studies focussing on organic matter dynamics showed a high microbial activity in topsoil from Rotthalmünster (e. g. John et al. (2005), Helfrich et al. (2006)). Thus we assume a high denitrification potential under $O_2$ depleted conditions. Denitrification activity was recently investigated and proved by Malique et al. (2019).*

*Although topsoil communities are exposed to $O_2$ enriched environments in general, temporal $O_2$ depletion also occurs following rain events or freeze/thaw cycles. Thus, under $O_2$ depleted conditions specialized microbial organisms are capable to switch from aerobic respiration to denitrification, which is a facultative anaerobic process. It is well known that in soil with $\geq$ 60% WFPS different processes co-occur, such as nitrification, nitrifier denitrification and denitrification. This is the reason why we assumed a denitrification potential in the topsoil of both soils that is dependent on the $O_2$ availability.*

*We added this point in the Material & Method section as follows (l. 117 ff.): "Fine-textured topsoil material was collected from two different agricultural sites in Germany (from a depth of 10 - 20 cm in Rotthalmünster (RM) and of 3 - 15 cm in Gießen (GI) as representatives for agricultural mid-European soils (Table 1). Malique et al. (2019) recently investigated the denitrification potential of both soils and found a higher denitrification activity with GI soil compared to that of RM soil."*

(2) Pls add few sentences for future study about microbial communities. The implicit assumption of this study is that similar community structure of microbiomes exist in physiochemically distinct soils, leading to similar responsive patterns under oxygen and substrate supply. This is also somehow astonishing. At least the flux pattern are largely similar, which may represent similar communities?

*The three different water saturations affect the microbial community directly, i. e. switching from aerobic to anaerobic respiration with high water saturation. Although we have no information on the microbial community structure in both soils we were aware that microbial groups had to adapt (i. e. expression of denitrification genes as a response to $O_2$ depletion) to changes in environmental conditions after raising the water saturation. This was the reason for excluding gas fluxes of the initial 24h of incubation as we accounted this as an equilibrium phase. However, as we have no information on this, we did not assume that the microbial community structure is similar in both soils, but assumed a relatively short-term expression of the respective genes for denitrification of facultative anaerobic organisms as a response of changing the $O_2$ conditions. Unfortunately, we could not assess the diversity of microorganisms involved in denitrification in the present study. However, we accounted the higher $CO_2$, $N_2O$ or $(N_2O+N_2)$ fluxes from GI soil compared to that from RM soil as differences in microbial activity and stated in l. 123 ff. that the "SOM content should mainly govern microbial activity and thus $O_2$ demand".*

*We inserted that microbial structure has to be taken into account in l. 604 ff. "Under field conditions this impact on denitrification is additionally altered by saturation changes, temperature variations, atmospheric gas concentrations, microbial community structure, and plant growth." As we have no information on the microbial community structure we provided this information in line 473 ff. as follows: "Other factors that would have affected $O_2$ demand and energy sources for denitrifiers (quality of organic matter, temperature, pH, plant-soil interactions), $O_2$ supply (oxygen concentration in the headspace, temperature) or other drivers of denitrification ($NO_3^-$ concentration, pH, denitrifier community structure) were either controlled or excluded in this study."*

(3) Why not use destructive sampling for microstructure analysis. Yes, one time point result cannot represent the entire incubation period, and why destructive sampling could be conducted to have a time-series analysis of microstructure and ansvf? For example, microbial activity could likely reach a high level after incubation for 24 hours, and stayed largely at a plateau level after incubation for 192 hours. In addition, strong respiration may lead to the growth of microorganisms which in turn generate extracellular enzymes or extracellular polymer substance EPS, which could likely significantly distort the soil microstructure and ansvf?

*Thank you for this comment. However, due to the experimental setup, it was only possible to scan the soil cores with X-ray CT once directly after the incubation experiment. The soil core was installed in a closed system, including flushing the headspace. Destructive sampling in-between the incubation was thus impossible as it would have affected the whole gas measurements and also the bulk soil mass of one soil core and this would affect the image analysis. Thus, parallel incubations would be the only option to sample microstructure or microbial community during the incubation period. Unfortunately, it was just impossible to run such a complex experimental setup with even more treatments during the project time of 3 years. In total we had 36 incubation vessels (2 soils x 2 aggregate sizes x 3 water saturations) that were incubated over 9 days and carried out in two working weeks. This resulted already in 72 weeks of incubations. Unfortunately, additional vessels to perform destructive sampling of parallel samples would have been too time consuming, although as you also argued it would have provided additional and important information. Please note that X-ray CT is not suitable to detect the release of enzymes and EPS and that it is very unlikely that those compounds would change the soil structure. If anything they would change structural stability, but there are no mechanical stresses during incubation for which aggregate stability would be relevant.*

(4) Can the authors specify the relationship between the volume of a single cell (or population) and the aggregate size. Or in other words, what is the population size of cell that can colonize different aggregates

*We did not perform any microbial or genetic based method. Watt et al. (2006) found $10^7$-$10^{12}$ of microbial cells in one gram soil. It is well known that colonization depends on substrate, oxygen and water availability, as summarized by Sessitsch et a. 2001. In the presented study we used macroaggregates (i. e. 2-4 and 4-8 mm), and both aggregate sizes were assumed to provide variations in $O_2$ supply for microorganisms inhabiting the aggregates. Juyal et al. (2018) performed experiments using sterilized repacked soils of two different aggregate sizes inoculated with two bacterial strains (*Bacillus sp. *and* Pseudomonas sp.*). They found higher cell numbers of* Bacillus sp. *in smaller aggregates (1-2mm) compared to the larger ones (2-4 mm) and attributed this to better nutrient availability in smaller aggregates. However, with the second bacteria* Pseudomonas sp. *they did not find such effects of aggregate sizes and this was in line with previous results by Drazkiewicz (1994). These results indicate that aggregates sizes affect growth of microbial species differently and consequently we cannot provide estimates for cell numbers under presented experimental conditions. Further experiments would be necessary to answer this question.*

*Although microbial growth of cell number within different aggregate sizes might be different, we did not find aggregate effects on denitrification and in l. 578 ff. we focussed on this point as follows: "The fact that aggregate size had no effect on denitrification indicates that critical distances were larger than the aggregate radii and rather controlled by air distribution in the macropore system. This means that both aggregate sizes used in the present study might have been too small to provoke differences in $CO_2$, $N_2O$ and ($N_2O+N_2$) fluxes. The large distance found here is in contrast to the very short critical distances of 180 µm for sufficient soil aeration estimated by Kravchenko et al. (2018) and Kravchenko et al. (2019) for intact soil*

*cores containing crop residues for which soil respiration was not determined but likely to be much higher."*

**The editor has some minor concerns**

**Comments to the Author:**

My only concern is that the title may be rephrased as Denitrification in soil as a function of oxygen availability at the microscale or Denitrification in soil as a function of oxygen supply at the microscale

> *Thank you for pointing out that the title can be shortened. As we focussed on factors controlling oxygen demand and oxygen supply, we decided to change the title to "Denitrification in soil as a function of oxygen availability at the microscale"*

In addition, please check the tables and figures. It is expected that your figure legends will be quite detailed and very precise. In fact, from the figure title and the axis labels of a graph/table the reader should be able to determine the question being asked, get a good idea of how the study was done, and be able to interpret the figure without reference to the text

> *We took great care to revise the figure and table titles and captions and precisely described the content of figures or tables.*

**The editor has some minor concerns**

(1) Effect of aggregate sizes on denitrification and $CO_2$ release is not significant. The authors argued about the smaller aggregate radii than the thresholds of distance connecting air aeration. Please add few sentences about the oxygen availability. Maybe the oxygen is completely depleted, or maybe oxygen is below threshold value that facilitates denitrification and soil respiration.

> *As described in the introduction (l. 49 f.), denitrification only occurs under anaerobic soil conditions, i.e. when oxygen is depleted locally. However, even a soil that still contains substantial amounts of oxygen on average can evoke denitrification in anaerobic microsites. We already discussed that the anaerobic soil volume fraction (ansvf) is a function of*

*connected air in soil ($O_2$ supply), and microbial activity (i.e. $O_2$ demand) may lead to local variations of the ansvf and this could also enhance development of anaerobic conditions and promote denitrification (l. 485 ff. and l. 504 ff.).*

*As mentioned by the editor, we argued that the critical distance to connected pores for an anaerobic microsite to emerge was larger (5 mm) than the aggregate radii (maximum of 2 or 4 mm, respectively). Thus, in the present study the aggregate radii were too small to affect $O_2$ availability and thus the anaerobic soil volume fraction at any of the investigated soil moistures. When air content was high, all visible macropores where air-filled so that this critical air distance was hardly exceeded anywhere. When air content was low (close to full water saturation), the patchy distribution of air and water in the macropore system was governed by subtle layering in the pore structure and not by aggregate size.*

*We rephrased this in line 513 ff. "The fact that aggregate size had no effect on denitrification indicates that critical distances were larger than the aggregate radii and rather controlled by air distribution in the macropore system. When air content was high, all visible macropores where air-filled so that this critical air distance was hardly exceeded anywhere. When air content was low (close to full water saturation), the patchy distribution of air and water in the macropore system was governed by subtle layering in the pore structure and not by aggregate size. This means that both aggregate sizes used in the present study might have been too small to provoke differences in $O_2$ availability and thus in $CO_2$, $N_2O$ and $(N_2O+N_2)$ fluxes."*

(2) Denitrification in upland soil. Please add few sentences about the flux of $N_2O$ emission in the field to justify the choice of soil samples.

*Both soils are representatives for agricultural mid-European soils (mentioned in l. 118) and have the potential for denitrification activity as recently investigated by Malique et al. (2019) in a laboratory experiment, what justifies the choice of both soils to perform the presented study. We referred on this in the Material & Methods section as follows (l. 118 ff.: "Malique et al. (2019) recently investigated the denitrification potential of both soils and found a higher denitrification activity with GI soil compared to that of RM soil. According to this, these soils were chosen for the contrast in properties potentially affecting denitrification and respiration (SOM contents, pH, texture, bulk density) which induces a large difference in microbial respiration and hence $O_2$ demand under identical incubation settings."*

*The literature about $N_2O$ fluxes in the field of the two soils tested in the presented study (from Gießen and Rotthalmünster in Germany) is limited. When investigating field fluxes, one must bear in mind, that under natural conditions several $N_2O$ producing pathways may co-occur in the soil, such as nitrification, denitrification or nitrifier denitrification, depending on the prevailing conditions. Additionally, $N_2O$ emissions do not directly reflect the denitrification activity since $N_2O$ can be further reduced to $N_2$. Unfortunately, to our knowledge, there are no data of field measurements of denitrification with the Rotthalmünster soil, but there are some studies that measured $N_2O$ fluxes in the field from the Gießen soil (Regan et al., 2011; Kammann et al., 2008; Müller et al., 2004). $N_2O$ emissions up to 45 µg $N_2O$-N $m^{-2}$ $h^{-1}$ were measured in an experiment in 2001 over 20 weeks (Müller et al. 2004) and up 15 µg $N_2O$-N $m^{-2}$ $h^{-1}$ were found in 2008 from the GI soil after fertilization (Regan et al. 2011). Müller et al. 2004 could show that a large amount of $N_2O$ emissions derived from $NO_3^-$ reduction.*

*We will change the above mentioned section (l. 118 ff.) by "To our knowledge, $N_2O$ field measurements only exist for GI soil which amounted to $N_2O$ emissions up to approximately 160 µg $N_2O$-N $m^{-2}$ $h^{-1}$ after fertilization (Müller et al., 2004; Kammann et al., 2008; Regan et al., 2011). Denitrification potential, however, exists in both soils, as recently investigated by*

*Malique et al. (2019) in a laboratory experiment with both soils. A higher denitrification activity with GI soil was found compared to that of RM soil (Malique et al. 2019). According to this, …".*

Meanwhile, I guess in future metatranscriptomic analysis could be of great help for such kind of study, in addition to destructive sampling of the paralleled microcosms.

*Thank you for pointing this out. As mentioned before, we agree that this information would be very interesting and helpful for interpretation of results. However, we have presented a very comprehensive experimental setup, combining gas flux measurements, isotopic analysis, and image analysis of CT derived data as well as simulating the diffusivity. These were very time consuming methods, especially the demanding image analysis. Methods to analyse the denitrifying communities in soil are not established in our lab and unfortunately we were not able to perform genetic analysis. In the revised version the microbial community was added to the other factors altering denitrification under field conditions in the section 4.3 "Future directions and implications for modelling" (l. 606 ff.): "Under field conditions this impact on denitrification is additionally altered by saturation changes, temperature variations, atmospheric gas concentrations, microbial community structure, and plant growth."*

[revised manuscript text omitted]

*Detailed information on pre-incubation, determination of water holding capacity and experimental set-up (Section 2: Material and Methods, 1. Incubation)*

For pre-incubation the soil was loosely placed on a tray, adjusted to 50% water holding capacity (WHC) with a spray can and stored at room temperature in the dark for two weeks.

Additional soil cores with the same dimension were packed in an identical manner as described in the Material and Method section and fully saturated by immersion in a water bath for 24h. The water-holding capacity (v/v % WHC) for each soil material was determined after free drainage. These water volumes were taken as a reference to adjust the above-mentioned saturation levels (70, 83 and 95% WHC). Note that WHC values are not identical to water saturations expressed in v/v% water-filled pore space (WFPS), since 100%WHC covers a smaller volume than the total pore volume due to 1) air entrapment during full immersion in water and 2) drainage of the biggest pores in a pressure head range of -10 to 0 cm in a 10 cm tall, freely draining sample.

The cylindrical PVC columns containing the packed soil aggregates (698.41 $cm^3$) were closed tightly by sealing caps at the top and bottom. The closed column was equipped with an in- and outlet to allow flushing the headspace (69.83 $cm^3$) through steel capillaries (total volume 1.33 $cm^3$). A maximal evaporation loss during incubation of one soil core is estimated to be around 1.22 g $H_2O$. A temperature sensor (PT100) was installed through the centre of the lid reaching the repacked aggregates with a depth of ca 3 cm down to assure constant temperature of 20°C during incubation.

*Table with average data for each treatment (WFPS and aggregate size) with average values of CO₂, N₂O and (N₂O+N₂) fluxes, O₂*
*saturation, total porosity, visible air content ($\varepsilon_{vis}$), connected air content ($\varepsilon_{con}$), anaerobic soil volume fraction (ansvf), simulated*
*diffusivity ($D_{sim}$) and product ratio (pr) for soil from Gießen (GI) and Rotthalmünster (RM)*

**Table S1: Average values for CO₂, N₂O and (N₂O+N₂) fluxes, O₂ saturation, visible air content ($\varepsilon_{vis}$), connected air content ($\varepsilon_{con}$), anaerobic soil volume fraction (ansvf),**
**simulated diffusivity ($D_{sim}$) and product ratio (pr) [N₂O/(N₂O+N₂)] for the two soils (Gießen (GI) and Rotthalmünster (RM)), three water saturations (water filled pore**
**space (WFPS)) and two aggregate sizes. Standard error (n=3) is shown in the brackets.**

| soil | WFPS [%] | Aggregate size [mm] | CO₂-C [µg h⁻¹ kg⁻¹] | N₂O-N [µg h⁻¹ kg⁻¹] | (N₂O+N₂)-N [µg h⁻¹ kg⁻¹] | O₂ [%air saturation] | Total porosity [-] | $\varepsilon_{vis}$ [-] | $\varepsilon_{con}$ [-] | ansvf [-] | $D_{sim}$ [m² s⁻¹] | pr [-] |
|---|---|---|---|---|---|---|---|---|---|---|---|---|
| GI | 63 | 2-4 | 535.71 (72.95) | 0.26 (0.07) | 2.94 (1.75) | 47.99 (1.30) | 0.21 (0.03) | 0.21 (0.03) | 0.20 (0.03) | <0.01 (<0.01) | 1.09 10⁻⁰⁶ (1.82 10⁻⁰⁸) | 0.34 (0.16) |
| GI | 63 | 4-8 | 503.19 (65.9) | 1.28 (0.67) | 2.93 (0.45) | 55.69 (1.87) | 0.20 (0.02) | 0.20 (0.02) | 0.20 (0.02) | <0.01 (<0.01) | 1.08 10⁻⁰⁶ (1.56 10⁻⁰⁸) | 0.44 (0.09) |
| GI | 75 | 2-4 | 617.30 (53.06) | 18.01 (3.00) | 35.53 (2.15) | 56.48 (2.50) | 0.18 (0.03) | 0.13 (0.03) | 0.12 (0.03) | 0.04 (0.02) | 1.59 10⁻⁰⁸ (7.26 10⁻⁰⁹) | 0.52 (0.08) |
| GI | 75 | 4-8 | 548.66 (57.25) | 17.89 (1.94) | 26.90 (4.42) | 61.78 (2.22) | 0.19 (0.03) | 0.14 (0.03) | 0.11 (0.04) | 0.21 (0.07) | 2.76 10⁻⁰⁹ (2.32 10⁻⁰⁹) | 0.68 (0.06) |
| GI | 85 | 2-4 | 175.33 (71.30) | 18.74 (7.51) | 27.20 (6.41) | 33.77 (1.47) | 0.18 (0.03) | 0.12 (0.02) | 0.03 (0.03) | 0.79 (0.14) | 5.59 10⁻¹⁰ (3.36 10⁻¹⁰) | 0.64 (0.09) |
| GI | 85 | 4-8 | 125.62 (21.69) | 13.30 (4.45) | 21.38 (1.97) | 39.89 (2.55) | 0.20 (0.03) | 0.10 (0.02) | 0.01 (0.02) | 0.80 (0.09) | 2.00 10⁻¹⁰ (4.00 10⁻¹¹) | 0.60 (0.10) |
| RM | 65 | 2-4 | 144.85 (20.45) | 0.02 (0.01) | NA | 55.11 (2.20) | 0.16 (0.03) | 0.16 (0.03) | 0.15 (0.03) | <0.01 (<0.01) | 2.24 10⁻⁰⁷ (1.39 10⁻⁰⁸) | n.d. |
| RM | 65 | 4-8 | 158.06 (21.05) | 0.05 (0.03) | 0.66 (0.54) | 48.95 (2.56) | 0.15 (0.03) | 0.15 (0.03) | 0.15 (0.03) | <0.01 (<0.01) | 2.08 10⁻⁰⁷ (2.69 10⁻⁰⁸) | 0.08 (0.04) |
| RM | 78 | 2-4 | 174.29 (4.14) | 4.28 (2.04) | 6.86 (3.28) | 59.16 (2.88) | 0.14 (0.03) | 0.10 (0.03) | 0.09 (0.03) | 0.08 (0.06) | 1.03 10⁻⁰⁸ (3.65 10⁻⁰⁹) | 0.65 (0.08) |
| RM | 78 | 4-8 | 142.69 (26.87) | 6.00 (1.18) | 9.88 (1.91) | 53.41 (2.60) | 0.14 (0.03) | 0.10 (0.03) | 0.07 (0.04) | 0.34 (0.22) | 1.47 10⁻⁰⁸ (7.34 10⁻⁰⁹) | 0.61 (0.05) |
| RM | 88 | 2-4 | 50.60 (7.49) | 5.07 (0.96) | 8.46 (2.48) | 22.61 (1.95) | 0.10 (0.02) | 0.06 (0.02) | 0.03 (0.02) | 0.69 (0.10) | 3.27 10⁻¹¹ (2.02 10⁻¹¹) | 0.64 (0.06) |
| RM | 88 | 4-8 | 46.89 (10.41) | 5.60 (1.15) | 8.50 (1.92) | 42.01 (2.59) | 0.13 (0.03) | 0.07 (0.02) | 0.02 (0.01) | 0.74 (0.07) | 2.03 10⁻⁰⁹ (1.76 10⁻⁰⁹) | 0.67 (0.04) |

n.d.: not detectable; NA: not applicable

*N₂O and CO₂ fluxes and O₂ saturation as a function of incubation time*

N₂O and CO₂ fluxes (Figure S1) and O₂ saturation at 7 locations within the soil core (Figure S2) were measured during the incubation time of approximately 192h. In the beginning of incubation establishment of equilibrium was assumed and therefore 24h of measurements in the beginning of the incubation time were excluded.

[Figure]

**Figure S1: Average N₂O and CO₂ fluxes as a function of incubation time for soil from Rotthalmünster (RM) in red and**
**Gießen (GI) in blue, two aggregate sizes (2-4 and 4-8 mm) and three water saturations (dotted, dashed or solid line**
**depicted lowest (63 or 65 % water filled pore space (WFPS) with GI and RM soil, respectively), medium (75 or 78 %**

**WFPS with GI and RM soil, respectively)** and highest **(85 or 88 % WFPS with GI and RM soil, respectively)** water
**saturation, respectively) with three replicates.**

**Figure S2: Average O$_2$ saturations measured by  seven sensors per soil core as a function of incubation time for soil from**
**Rotthalmünster (RM) in red and Gießen (GI) in blue, two aggregate sizes (2-4 and 4-8 mm (solid and dashed lines,**
**respectively)) and three  water filled pore spaces (WFPS) with three replicates each. Only the final 24h**
**were considered for regression analysis N-gas release and X-ray CT results.**

*Detailed description of calculating different pools for $^{15}N$*

The fraction of N in N$_2$O ($f_p\_N_2O$) or N$_2$ ($f_p\_N_2$) originating from $^{15}$N-labelled NO$_3^-$ pool within one sample was calculated according to (Spott et al., 2006; Lewicka-Szczebak et al., 2013; Well et al., 2019) using the $^{15}$N abundance of N$_2$ or N$_2$O measured in the analyzed gas sample ($a_m$), in the non-labelled N$_2$ in technical gas ($a_{bgd}$), and the calculated $^{15}$N abundance of the active NO$_3^-$ pool ($a_p$).

$$f_p\_N_2O \;=\; \frac{a_m - a_{bgd}}{a_p - a_{bgd}} \tag{1}$$

$$f_p\_N_2 \;=\; \frac{a_m - a_{bgd}}{a_p - a_{bgd}} \tag{2}$$

with

$$a_m = \frac{{}^{29}R + 2\,{}^{30}R}{2(1 + {}^{29}R + {}^{30}R)} \tag{3}$$

and using the fraction of $^{30}$N$_2$ in the gas sample ($^{30}\chi_m$):

$$a_p = \frac{^{30}\chi_m - a_m \cdot a_{bgd}}{a_m - a_{bgd}}$$ (4)

This is based on the a non-random distribution of isotopes in $N_2O$ and $N_2$ (Spott et al., 2006):

$$^{30}\chi_m = \frac{^{30}R}{1 + {}^{29}R + {}^{30}R}$$ (5)

Thus, with $f_p\_N_2O$ the $N_2O$ flux from denitrification ($N_2O\_deni$) was calculated

$$N_2O\_deni = N_2O\_total * f_p\_N_2O$$ (6)

The $f_p\_N_2O$ was constantly near 1 for both soils, aggregate sizes, water saturations and time points of sampling resulting in very similar $N_2O\_total$ and $N_2O\_deni$ values (Figure S3). The time resolution for

$N_2O\_total$ was much higher than for isotopic analysis and therefore $N_2O\_total$ was used to calculate $N_2O$

fluxes from denitrification and for statistical analysis.

[Figure]

**Figure S3: Comparison of total $N_2O$ emissions ($N_2O\_total$) captured by gas chromatography and $N_2O$ emissions from**
**denitrification ($N_2O\_deni$) calculated by Eq. 6 from experimental treatments with soil from Rotthalmünster (RM) and**
**Gießen (GI), two aggregate sizes (2-4 and 4-8 mm) and three water saturations. Goodness of fit to the 1:1 line (gray line) is**
**expressed as slope and $R^2$ from linear regression. The excellent agreement implies that $N_2O$ is produced by**
**denitrification.**

*Impact of packing procedure on visible air content ($\varepsilon_{vis}$) and anaerobic soil volume fraction*

*(ansvf)*

[Figure]

**Figure S4: Visible air content ($\varepsilon_{vis}$) and the anaerobic soil volume fraction (*ansvf*) as a function of soil core depth for soil**
**from (a) Gießen (GI) and (b) Rotthalmünster (RM). Shown here are examples of  three replicates of repacked soil cores**
**with aggregates of 4-8 mm size incubated at medium water saturation of 75% with GI and 78% with RM soil. Values**
**shown here for $\varepsilon_{vis}$ air content and *ansvf*  are aggregated for 4.7 mm segments in depth.**
**The results show that the *ansvf* increases substantially in thin soil layers with low $\varepsilon_{vis}$ created by packing in which air**
**continuity is lost.**

Two representative examples of one treatment were chosen to illustrate the impact of packing the soil on visible air content ($\varepsilon_{vis}$) and anaerobic soil volume fraction (*ansvf*) (large aggregates of GI soil incubated at 75% WFPS and large aggregates of RM soil incubated at 78 % WFPS) (Figure S4). During the packing procedure, intervals of 2 cm were the best option to adjust the target material-specific bulk densities and water saturations within the soil core. The average $\varepsilon_{vis}$ did not differ between replicates of one treatment (Figure 4), but decreased with increasing depth of the packed soil core and was extremely reduced at the top of one packing interval (Figure S4). This varying compaction in different layers affected also the *ansvf* of each repacked core (Figure S4). The *ansvf* dramatically increased in layers, where lowest $\varepsilon_{vis}$ was observed. In some cases, the *ansvf* even reached 1, i.e. complete exclusion from connected air-filled pores.

*Detailed information on simulated diffusivity ($D_{sim}$)*

Diffusivity was simulated for individual aggregates as well as for the entire soil core (bulk diffusivity)

directly on segmented X-ray CT data on a workstation with Intel® Xeon® CPUs (E7-8867v4, 2.46Hz, 36

cores) and 6.1TB RAM by solving the Laplace equation with the DiffuDict module in the GeoDict 2019

Software (Math2Market GmbH, Kaiserslautern, Germany). A hierarchical approach was used to estimate the effective diffusivity of the wet soil matrix by simulating Laplace diffusion on cubes contained in individual soil aggregates with the Explicit Jump solver assuming free diffusion in the visible pore space, a completely impermeable background and symmetric boundary condition on all sides (Wiegmann and

Zemitis, 2006; Wiegmann and Bube, 2000). The resulting effective diffusion coefficient is expressed as a percentage of the diffusion coefficient in the free fluid and was in the range of $6.6 \times 10^{-4} \pm 3.7 \times 10^{-4}$% and 2.4

$10^{-2} \pm 1.3 \times 10^{-2}$% for wet aggregates of RM and GI soil, respectively. For the soil cores with <70% WFPS

the visible pore space in the high-resolution aggregate images is assumed to be air-filled, whereas for soil cores with ≥75% WFPS it is assumed to be water-filled, which is justified by the fact that 1) the air-filled porosity at <70% WFPS in individual aggregates (RM: 17.6%, GI: 23.1%) exceeds the visible pore space in low-resolution soil core images (RM: 15.8%, GI: 20.6%) and 2) that in contrast to the higher moisture levels no free water could be identified at the column scale with air-filled porosity at <70% WFPS. Thus, the effective diffusion coefficient for soil matrix is determined with respect to the oxygen diffusion coefficient ($D_{O2}$) at 2% $O_2$ in pure air ($2.03 \times 10^{-5}$ m² s$^{-1}$) and in pure water ($1.97 \times 10^{-9}$ m² s$^{-1}$) at 20°C, respectively (http://compost.css.cornell.edu/oxygen/oxygen.diff.air.html).

Another series of diffusion experiments was modeled with the Explicit Jump solver on the entire soil cores (1550x1550x [1500-1600] voxels) with the effective diffusion coefficient of the soil matrix taken from aggregate simulations, an impermeable exterior, impermeable mineral grains (GI only) and the diffusion coefficient of oxygen in air and water (≥70% WFPS only) in the respective material classes. In order to save memory, periodic boundary conditions were assumed on all sides. This is irrelevant for lateral boundaries as they are blocked by the impermeable exterior anyway, but may lead to a lower effective diffusion coefficient, since the spatial distribution of materials at the top and bottom of the domain do not match, which imposes an additional diffusion barrier. The reduction by this discontinuity was in the range of $5.1 \times 10^{-9}$ to $6.7 \times 10^{-8}$ m² s$^{-1}$ in small test images (500³ voxels) from all soil materials and saturations.

*Product ratio (pr) as a function of time*

[Figure]

.

**Figure S5:** Product ratio (*pr*) [N$_2$O/(N$_2$O+N$_2$)]  as a function of time for soil from Gießen (GI) in blue and Rotthalmünster (RM) in red with aggregates of 2-4 mm and 4-8mm size incubated at three water filled pore spaces (WFPS). The lines connect the average values of three replicates (large and small aggregates, respectively). The *pr* decreases gradually over time due to a relative increase in N$_2$.

*Correlation between ansvf and gas emissions and concentrations and emissions*

The correlation of *ansvf* with average gas fluxes and internal $O_2$ concentrations is shown in Figure

5S6. Since the drop in $CO_2$ release at the highest water saturations coincided with an escalating *ansvf*, the relation between the two was highly correlated (Spearman's $R$>-0.7 and p=0.04) for all soils and aggregate sizes (Figure 5S6a), but with different slopes for both soils due to vastly different SOM

contents. The correlation of *ansvf* with $N_2O$ is weaker (Spearman's 0.6<R<0.77) and on the verge of being significant (p≤0.1) (Figure 5S6c). However, the correlation of *ansvf* with $(N_2O+N_2)$ release is even worse (p>0.2), so the mechanisms that govern $N_2O$ and $(N_2O+N_2)$ release must be more complex (Figure

5S6c, d). As expected the average $O_2$ saturation decreases with increasing *ansvf* (Figure 5S6b). Yet, correlation is lower than for $CO_2$ (Spearman's -0.6<R<-0.2, but p>0.2), likely due to limited representativeness of average $O_2$ concentrations derived from a few point measurements.

[Figure]

**Figure S6: Average (a) $CO_2$, fluxes (b) $O_2$ saturation, (c) $N_2O$ and (d) $(N_2O+N_2)$ fluxes as a function of anaerobic soil**
**volume fraction (*ansvf*) for soil from Rotthalmünster (RM) and Gießen (GI) and two aggregate sizes (2-4 and 4-8 mm) for**
**three individual replicates. The Spearman's rank correlation coefficient (*R*) result from Spearman's rank correlation and**
**indicate the extent of monotonic relation between the ranks of both variables. The associated p-values (*p*) were corrected**
**for multiple comparison according to Benjamini and Hochberg (1995).**

*Correlation matrix between all variables*

[Figure]

**Figure S7: Correlation matrix of Spearman's rank correlation showing coefficients (*R*) between two measured**
**variables ($N_2O$, ($N_2O+N_2$) or $CO_2$ fluxes, anaerobic soil volume fraction (*ansvf*), product ratio (*pr*), $O_2$ saturation ($O_2$),**
**simulated diffusivity ($D_{sim}$) or connected air content ($\varepsilon_{con}$)) in one cell with pairwise deletion of missing values. Asterisks**
**indicate the statistical significance with significance levels of *p ≤ 0.05, **p ≤ 0.005, ***p ≤ 0.001 for adjusted p-values**
**according to the method of Benjamini and Hochberg (1995). Color scheme indicate low (light colors) or strong (intensive**
**colors) correlation as well as positive (red) or negative (blue) correlation.**

| *Explanatory variables for denitrification*

[Figure]

**Figure S8: Biplot of the PLSR results for response variables $N_2O$ (a) and ($N_2O+N_2$) fluxes (b) showing x-scores and x-loadings of two components (Comp 1 and Comp 2). The x- and y- axis represent values of the scores for soil from Gießen (GI) in blue and Rotthalmünster (RM) in red with aggregates of 2-4 mm (triangles) and 4-8 mm size (circles) incubated at three water saturations depicted by the size of symbols. The second y-axis represents values for the loadings (predictors and arrows) to show the influence of variables on the components.**

The regression equations with $R^2$ values and a confidence interval of 95% in square brackets resulting from PLSR with $CO_2$, (*pr*) and *ansvf* as explanatory variables to predict $N_2O$ or ($N_2O+N_2$) fluxes of the present study for data after log- or logit transformation:

$$\log(N_2O) = 0.63\log(CO_2) + 0.41\operatorname{logit}(ansvf) + 0.64\,pr - 0.38\log(D_{sim}) - 0.22\,\varepsilon_{con} + 0.12\,O_2;$$

$$R^2 = 0.\underline{82}\,[0.\underline{65}\text{-}0.\underline{91}] \tag{7}$$

$$\log(N_2O + N_2) = \underline{0.}1.1\log(CO_2) + 0.70\operatorname{logit}(ansvf) - 0.65\log(D_{sim}) - 0.37\,\varepsilon_{con} + 0.10\,O_2\;;$$

$$R^2 = 0.\underline{78}\,[0.\underline{62}\text{-}0.\underline{85}] \tag{8}$$

*Empirical models to calculate the diffusivity of the soil cores*

It is assumed, that the total porosity (Φ) was unaffected by the packing procedure, whereas the air content (ε) is expected to differ from the theoretic value due to compact regions and intervals caused by the packing (Figure S4). Following from this, the target bulk density of the repacked soil cores was used to calculate Φ (0.62 or 0.51 for GI and RM soil, respectively), while CT-derived ε was used. This enabled to calculate diffusivity based on the frequently used model of Millington and Quirk (1960), Millington and Quirk (1961), Moldrup et al. (2000) and also according to the model of Deepagoda et al. (2011) (Table S2, Figure S9). As expected, diffusivity from these models has a lower explanatory power for $N_2O$ and $(N_2O+N_2)$ release compared to $D_{sim}$ of the present study (3D simulation) (Table S2). Higher diffusivities for treatments ≥75% WFPS from empirical models ($D_{emp}$) compared to $D_{sim}$ result from heterogeneities in compaction of the repacked soil core as described earlier ( Figure S4, Figure S9), while empirical models were developed for natural soils that very likely possess higher air continuity at low air content. These empirical models only take averages for porosity and water-filled pores into account (Millington and Quirk, 1961; Moldrup et al., 2000) (Figure S9, Table S2), whereas heterogeneities in compaction are explicitly considered in 3D diffusivity simulations ($D_{sim}$).

[Figure]

**Figure S9: Simulated diffusivities ($D_{sim}$) of the present study (blue circle) and calculated diffusivities as a function of WFPS for both soils (Rotthalmünster (RM) and Gießen (GI)). Models used to calculate diffusivity are published by Millington and Quirk (1960) (MQ_1960, green circle), Millington and Quirk (1961) (MQ_1961, light green circle), Moldrup et al. (2000) (Mol_2000, red circle) and Deepagoda et al. (2011) (DC_GMP_2011, purple circle). According to the calculations of the present study diffusivity in free air ($D_0$) was assumed to be 2.03 $10^{-5}$ $m^2$ $s^{-1}$.**

**Table S2: Explained variability (expressed as $R^2$ _for response variables $N_2O$ and $(N_2O+N_2)$_ with confidence interval of 95% in square brackets for $N_2O$ and $(N_2O+N_2)$ release obtained from partial least square regression (PLSR) using explanatory variables $CO_2$, diffusivity (and product ratio ($pr$) for $N_2O$ as response variable only). This was done to assess possibilities to substitute one of the most important explanatory variables ($ansvf$) by diffusivity. Data were pooled for both soils (RM and GI), WFPS treatments and aggregate sizes (n= 36). Diffusivity was obtained by 3D simulation of the present study ($D_{sim}$) or existing soil gas diffusivity models were used to calculate diffusivity, using total porosity ($\Phi$) and air content ($\varepsilon$) while diffusivity in free air ($D_0$) is assumed to be $2.03 \cdot 10^{-5}$ $m^2$ $s^{-1}$.**

| method | Equation to calculate diffusivity $D_{emp}$ [$m^2$ $s^{-1}$] | $R^2$ with response variable $N_2O$ | $R^2$ with response variable $(N_2O+N_2)$ |
|---|---|---|---|
| Present study[1] | $D_{sim}$ | 0.59 [0.34-0.78] | 0.63 [0.39-0.78] |
| Millington & Quirk (1961)[1] | $(\varepsilon^{10/3}/\Phi^2)$ $D_0$ | 0.46 [0.20-0.69] | 0.57 [0.28-0.78] |
| Millington & Quirk (1960)[1] | $(\varepsilon^2/\Phi^{2/3})$ $D_0$ | 0.48 [0.22-0.70] | 0.52 [0.21-0.74] |
| Moldrup et al. (2000)[1] | $\varepsilon^{1.5}$ $(\varepsilon/\Phi)$ $D_0$ | 0.59 [0.29-0.79] | 0.54 [0.24-0.75] |
| Deepagoda et al (2011)[1] | $0.1[2(\varepsilon/\Phi)^3+0.04(\varepsilon/\Phi)]$ $D_0$ | 0.52 [0.27-0.73] | 0.69 [0.42-0.82] |
| theoretic air content[2] | $\varepsilon_t$ | 0.55 [0.30-0.76] | 0.78 [0.57-0.90] |
| no diffusivity[3] | - | 0.48 [0.16-0.71] | 0.07 |

[1]**PLSR with $CO_2$ and diffusivity (and product ratio ($pr$)) as explanatory variables and $N_2O$ or $(N_2O+N_2)$ as response variables.**

[2]**Diffusivity substituted by the theoretic air content ($\varepsilon_t$) targeted during packing in PLSR _resulting in $CO_2$ and $\varepsilon_t$ (and product ratio ($pr$)) as explanatory variable for $N_2O$ and for $(N_2O+N_2)$_.**

[3]**Diffusivity was excluded in PLSR resulting in $CO_2$ (and product ratio ($pr$)) as explanatory variable for $N_2O$ and for $(N_2O+N_2)$. Because $CO_2$ was the single explanatory variable for $(N_2O+N_2)$ a simple linear model was used to estimate $R^2$.**

*Calculation of anaerobic soil volume fraction (ansvf) by (N₂O+N₂) fluxes from oxic and anoxic incubations*

To calculate an anaerobic soil volume fraction within the soil cores ($ansvf_{cal}$) independently from the X-ray CT imaging derived *ansvf,* parallel oxic and anoxic incubations were conducted using a different suite of larger repacked soil cores. The conditions for incubations were very similar in soil cores as described before (in the Methods section and Supplementary Material) for oxic incubation. Deviations from the experimental protocol were the dimension of the soil core (10x14.4 cm), unspecific sieving (>10 mm), a flow rate of 20 mL/min and a target saturation of 75% WFPS for both soils (GI and RM). Soil material for all incubations was obtained from the same batches. that had been used for the oxic incubations. Batches consisted of approx. 2000kg sieved, homogenized and air-dried soil stored at 6°C that had been collected and prepared to allow the study of comparable soil samples in various labs during several years. After one week withthree weeks with oxic incubation using a technical gas (20% $O_2$ and 2% $N_2$ in pure He) the atmospheric conditions were switched to anoxic conditions (2% $N_2$ in pure He). $N_2O$ and $N_2$ fluxes were quantified using the $^{15}N$ labelling approach as described before. A comparison of oxic and anoxic $(N_2O+N_2)$ fluxes under these comparable conditions is possible because $ansvf_{cal}$ assumes that actual denitrification is linearly related to *ansvf* and that the specific anoxic denitrification rate is homogenous, i.e. would be identical at any location within the soil.

The calculated *ansvf* (*ansvf$_{cal}$*) derived from incubation ($N_2O+N_2$) fluxes with oxic (*($N_2O+N_2)_{oxic}$*) and anoxic (*($N_2O+N_2)_{anoxic}$*) conditions is thus (Table S3):

$$ansvf_{cal} = \frac{(N_2O+N_2)_{oxic}}{(N_2O+N_2)_{anoxic}}$$                    (9)

**Table S3: Average ($N_2O+N_2$) fluxes with oxic conditions (*($N_2O+N_2)_{oxic}$* and with anoxic**
**conditions (*($N_2O+N_2)_{anoxic}$* (n=4) from parallel incubations for soils from Rotthalmünster (RM) and**
**Gießen (GI). **
** Anoxic conditions were established after 3 weeks of oxic incubation. Average ($N_2O+N_2$) fluxes from**
**oxic and anoxic incubations served to calculate the anaerobic soil volume fraction (*ansvf$_{cal}$*) (Eq. 9). In comparison to the**
**, *ansvf* derived from X-Ray CT imaging  from the present study is also presented.**

| soil | WFPS | Aggregate size [mm] | $(N_2O+N_2)_{oxic}$ [µg N h$^{-1}$ kg$^{-1}$] (parallel incubation) | $(N_2O+N_2)_{anoxic}$ [µg N h$^{-1}$ kg$^{-1}$] (parallel incubation) | *ansvf$_{cal}$* | *ansvf* (present study) |
|---|---|---|---|---|---|---|
| RM | 75-78 | 2-8 | 14±10 | 60±2 | 0.24±0.16 | 0.21 |
| GI | 75 | 2-8 | 61±97 | 136±18 | 0.45±0.71 | 0.13 |

*Table with data for each replicate with average values of $CO_2$, $N_2O$ and $(N_2O+N_2)$ fluxes, $O_2$ saturation, total porosity, visible air*
*content, connected air content ($\varepsilon_{con}$), anaerobic soil volume fraction (ansvf), diffusivity ($D_{sim}$) and product ratio (pr)*

**Table S4: Average values of $CO_2$, $N_2O$ and $(N_2O+N_2)$ fluxes, $O_2$ saturation, total porosity, visible air content ($\varepsilon_{vis}$), connected air content ($\varepsilon_{con}$), anaerobic soil volume**
**fraction (*ansvf*), diffusivity ($D_{sim}$) and product ratio (*pr*, [$N_2O/(N_2O+N_2)$]) for the two soils (Gießen (GI) and Rotthalmünster (RM)), three water saturations and two**
**aggregate sizes for three replicates. Standard error of the mean is shown in the brackets.**

| soil | WFPS [%] | Aggregate size [mm] | Replicate | CO$_2$-C [µg h$^{-1}$ kg$^{-1}$] (n=28) | N$_2$O-N [µg h$^{-1}$ kg$^{-1}$] (n=28) | (N$_2$O+N$_2$) [µg N h$^{-1}$ kg$^{-1}$] (n=3) | O$_2$ [%air saturation] (n=7) | Total porosity [-] | $\varepsilon_{vis}$ [-] | $\varepsilon_{con}$ [-] | ansvf [-] | D$_{sim}$ [m$^2$ s$^{-2}$] | pr (n= 1-3) |
|---|---|---|---|---|---|---|---|---|---|---|---|---|---|
| GI | 63 | 2-4 | a | 406.30 (3.24) | 0.22 (<0.01) | NA | 47.19 (12.13) | 0.20 | 0.20 | 0.19 | 0.003 | 1.10 10$^{-06}$ | n.d. |
| GI | 63 | 4-8 | a | 387.38 (2.83) | 0.52 (0.07) | 2.36 (NA) | 53.79 (13.07) | 0.19 | 0.19 | 0.19 | 0.004 | 1.05 10$^{-06}$ | 0.22 (n.d) |
| GI | 75 | 2-4 | a | 528.74 (3.73) | 21.28 (0.84) | 31.45 (7.65) | 46.27 (11.64) | 0.18 | 0.13 | 0.12 | 0.037 | 2.89 10$^{-08}$ | 0.68 (0.14) |
| GI | 75 | 4-8 | a | 463.32 (3.42) | 20.14 (0.60) | 30.21 (5.65) | 59.24 (11.59) | 0.19 | 0.14 | 0.10 | 0.246 | 7.50 10$^{-10}$ | 0.67 (0.12) |
| GI | 85 | 2-4 | a | 317.57 (2.55) | 33.68 (0.76) | 39.78(3.94) | 39.43 (9.42) | 0.17 | 0.11 | 0.07 | 0.513 | 1.54 10$^{-10}$ | 0.85 (0.06) |
| GI | 85 | 4-8 | a | 168.18 (2.30) | 22.11 (0.59) | 25.03 (2.79) | 39.66 (12.20) | 0.18 | 0.08 | 0.02 | 0.824 | 1.40 10$^{-10}$ | 0.88 (0.07) |
| GI | 63 | 2-4 | b | 542.08 (8.62) | 0.15 (<0.01) | 5.09 (NA) | 45.32 (10.48) | 0.22 | 0.22 | 0.21 | 0.001 | 1.11 10$^{-06}$ | 0.03 (n.d.) |
| GI | 63 | 4-8 | b | 506.33 (7.33) | 0.71 (0.01) | 2.62 (0.33) | 57.38 (11.56) | 0.21 | 0.21 | 0.21 | 0.001 | 1.11 10$^{-06}$ | 0.27 (0.03) |
| GI | 75 | 2-4 | b | 610.95 (4.95) | 20.73 (0.98) | 36.37 (10.48) | 62.33 (6.19) | 0.18 | 0.13 | 0.12 | 0.068 | 1.49 10$^{-08}$ | 0.57 (0.14) |
| GI | 75 | 4-8 | b | 525.22 (4.49) | 19.51 (0.83) | 32.34 (7.77) | 71.78 (7.66) | 0.19 | 0.14 | 0.10 | 0.312 | 1.52 10$^{-10}$ | 0.60 (0.12) |
| GI | 85 | 2-4 | b | 95.47 (3.03) | 12.48 (0.46) | 22.98 (7.01) | 28.45 (10.02) | 0.18 | 0.12 | <0.01 | 0.935 | 1.23 10$^{-09}$ | 0.54 (0.15) |
| GI | 85 | 4-8 | b | 97.08 (2.71) | 9.99(0.72) | 20.82 (9.16) | 34.16 (9.45) | 0.18 | 0.11 | <0.01 | 0.938 | 1.82 10$^{-10}$ | 0.48 (0.18) |
| GI | 63 | 2-4 | c | 658.77 (5.38) | 0.40 (0.01) | 0.80 (0.10) | 51.43 (9.55) | 0.21 | 0.21 | 0.20 | <0.001 | 1.05 10$^{-06}$ | 0.50 (0.04) |
| GI | 63 | 4-8 | c | 615.87 (4.61) | 2.63 (0.22) | 3.81 (1.00) | 70.19 (6.95) | 0.20 | 0.20 | 0.20 | <0.001 | 1.08 10$^{-06}$ | 0.69 (0.02) |
| GI | 75 | 2-4 | c | 712.21 (5.89) | 12.02 (0.90) | 38.77 (10.84) | 60.83 (8.62) | 0.19 | 0.13 | 0.13 | 0.018 | 3.88 10$^{-09}$ | 0.31 (0.05) |
| GI | 75 | 4-8 | c | 657.43 (5.30) | 14.03 (1.07) | 18.15 (4.37) | 54.30 (14.00) | 0.19 | 0.14 | 0.13 | 0.063 | 7.38 10$^{-09}$ | 0.77 (0.05) |
| GI | 85 | 2-4 | c | 112.95 (7.61) | 10.04 (1.16) | 18.83 (9.96) | 23.67 (10.43) | 0.18 | 0.12 | <0.01 | 0.910 | 2.98 10$^{-10}$ | 0.53 (0.21) |
| GI | 85 | 4-8 | c | 111.59 (6.66) | 7.80 (1.10) | 18.29 18.87) | 45.84 (10.25) | 0.23 | 0.12 | 0.02 | 0.629 | 2.75 10$^{-10}$ | 0.43 (0.18) |
| RM | 65 | 2-4 | a | 137.89 (0.65) | 0.01 (n.d.) | NA | 68.61 (7.14) | 0.15 | 0.15 | 0.14 | 0.004 | 2.51 10$^{-07}$ | n.d. |
| RM | 65 | 4-8 | a | 164.47 (0.90) | 0.10 (<0.01) | NA | 35.75 (12.64) | 0.16 | 0.16 | 0.15 | 0.005 | 2.47 10$^{-07}$ | n.d. |
| RM | 78 | 2-4 | a | 180.88 (1.57) | 0.22 (0.01) | 0.31 (0.10) | 63.18 (10.22) | 0.14 | 0.11 | 0.10 | 0.004 | 1.66 10$^{-08}$ | 0.71 (0.16) |
| RM | 78 | 4-8 | a | 71.12 (1.00) | 3.65 (0.21) | 6.11 (1.32) | 43.27 (11.97) | 0.14 | 0.08 | 0.03 | 0.775 | 2.34 10$^{-11}$ | 0.60 (0.06) |
| RM | 88 | 2-4 | a | 43.12 (0.19) | 3.27 (0.11) | 4.21 (0.73) | 12.13 (8.11) | 0.10 | 0.07 | 0.05 | 0.502 | 7.31 10$^{-11}$ | 0.78 (0.11) |
| RM | 88 | 4-8 | a | 26.20 (0.12) | 3.36 (0.08) | 4.83 (0.48) | 38.36 (11.27) | 0.10 | 0.05 | 0.02 | 0.753 | 5.53 10$^{-09}$ | 0.70 (0.04) |
| RM | 65 | 2-4 | b | 113.43 (0.75) | 0.04 (<0.01) | NA | 48.38 (11.00) | 0.17 | 0.17 | 0.16 | 0.003 | 2.10 10$^{-07}$ | n.d. |
| RM | 65 | 4-8 | b | 118.83 (0.85) | 0.05 (<0.01) | 1.31 (NA) | 42.40 (11.85) | 0.15 | 0.15 | 0.14 | 0.005 | 1.57 10$^{-07}$ | 0.04 (n.d.) |
| RM | 78 | 2-4 | b | 166.66 (1.95) | 6.12 (0.30) | 10.14 (3.34) | 56.52 (8.62) | 0.13 | 0.10 | 0.08 | 0.042 | 1.02 10$^{-08}$ | 0.60 (0.17) |
| RM | 78 | 4-8 | b | 163.13 (0.92) | 7.31 (0.19) | 11.25 (1.98) | 69.43 (9.15) | 0.14 | 0.11 | 0.09 | 0.193 | 2.13 10$^{-08}$ | 0.65 (0.10) |
| RM | 88 | 2-4 | b | 43.09 (0.20) | 5.43 (0.09) | 8.39 (1.01) | 28.13 (9.56) | 0.09 | 0.07 | 0.01 | 0.856 | 1.04 10$^{-11}$ | 0.64 (0.07) |
| RM | 88 | 4-8 | b | 55.12 (0.70) | 7.16 (0.16) | 11.30 (1.74) | 46.26 (9.60) | 0.14 | 0.07 | 0.01 | 0.860 | 3.65 10$^{-11}$ | 0.63 (0.09) |

| | | | | | | | | | | | | |
|---|---|---|---|---|---|---|---|---|---|---|---|---|
| RM | 65 | 2-4 | c | 183.25 (0.70) | n.d. | NA | 53.25 (14.68) | 0.17 | 0.17 | 0.16 | 0.003 | 2.10 10$^{-07}$ | n.d. |
| RM | 65 | 4-8 | c | 190.89 (0.82) | n.d. | NA | 68.71 (15.40) | 0.16 | 0.16 | 0.15 | 0.003 | 2.19 10$^{-07}$ | 0.11 (n.d.) |
| RM | 78 | 2-4 | c | 175.34 (0.30) | 6.51 (0.18) | 10.12 (2.29) | 57.79 (6.92) | 0.14 | 0.11 | 0.08 | 0.203 | 4.00 10$^{-09}$ | 0.64 (0.13) |
| RM | 78 | 4-8 | c | 193.83 (1.27) | 7.04 (0.46) | 12.29 (3.66) | 58.57 (12.57) | 0.14 | 0.11 | 0.10 | 0.062 | 2.28 10$^{-08}$ | 0.57 (0.12) |
| RM | 88 | 2-4 | c | 65.58 (0.40) | 6.53 (0.07) | 12.80 (2.94) | 27.69 (8.80) | 0.11 | 0.05 | 0.02 | 0.720 | 1.45 10$^{-11}$ | 0.51 (0.12) |
| RM | 88 | 4-8 | c | 549.33 (0.22) | 6.27 (0.12) | 9.36 (1.24) | 41.41 (9.23) | 0.16 | 0.08 | 0.03 | 0.613 | 5.19 10$^{-10}$ | 0.67 (0.08) |

n.d.: not detectable; NO and $N_2$ concentration was below detection limit for IRMS analysis, thus calculation of *pr* was impossible. NA: not applicable

Wiegmann, A., and Zemitis, A.: EJ-HEAT: A fast explicit jump harmonic averaging solver for the effective heat conductivity of composite materials, Berichte des Fraunhofer ITWM, 94, 2006.

